Corrected: Publisher Correction

# PRKCSH contributes to tumorigenesis by selective boosting of IRE1 signaling pathway

Gu-Choul Shin [1], Sung Ung Moon[2], Hong Seok Kang[1], Hyo-Sun Choi[1], Hee Dong Han[3] & Kyun-Hwan Kim [1,4]

Unfolded protein response (UPR) is an adaptive mechanism that aims at restoring ER homeostasis under severe environmental stress. Malignant cells are resistant to environmental stress, which is largely due to an activated UPR. However, the molecular mechanisms by which different UPR branches are selectively controlled in tumor cells are not clearly understood. Here, we provide evidence that PRKCSH, previously known as glucosidase II beta subunit, functions as a regulator for selective activation of the IRE1α branch of UPR. PRKCSH boosts ER stress–mediated autophosphorylation and oligomerization of IRE1α through mutual interaction. PRKCSH contributes to the induction of tumor-promoting factors and to tumor resistance to ER stress. Increased levels of PRKCSH in various tumor tissues are positively correlated with the expression of XBP1-target genes. Taken together, our data provide a molecular rationale for selective activation of the IRE1α branch in tumors and adaptation of tumor cells to severe environmental stress.

[1] Department of Pharmacology, Center for Cancer Research and Diagnostic Medicine, IBST, School of Medicine, Konkuk University, 1 Hwayang-Dong, Kwangjin-Gu, Seoul 143-701, Republic of Korea. [2] Center for Advanced Bioinformatics & Systems Medicine, Sookmyung Women's University, Hyochangwon-Gil 52, Youngsan-Gu, Seoul 140-741, Republic of Korea. [3] Department of Immunology, School of Medicine, Konkuk University, Chungwon-Daero 268, Chungju, Republic of Korea. [4] Research Institute of Medical Sciences, Konkuk University, 1 Hwayang-Dong, Kwangjin-Gu, Seoul 143-701, Republic of Korea. Correspondence and requests for materials should be addressed to G.-C.S. (email: rnelr@naver.com) or to K.-H.K. (email: khkim10@kku.ac.kr)

Cancer development is often associated with cytotoxic conditions such as nutrient deprivation, oxidative stress, and metabolic changes. These conditions trigger the unfolded protein response (UPR) that helps the cell to cope with the stress and to reestablish normal endoplasmic reticulum (ER) function by attenuating translation and promoting protein folding, secretion, and degradation[1–4]. Although prolonged ER stress can result in apoptotic cell death, chronic ER stress responses have been documented in most major types of human tumors and play a crucial role in tumor growth and survival[1–4]. UPR is also involved in tumor development and progression by promoting the expression of tumor growth factors such as TNF-α, IL-8, and VEGF[5,6]. UPR signaling is independently mediated by three distinct components: RNA-dependent protein kinase-like ER kinase (PERK), activating transcription factor 6 (ATF6), and inositol-requiring enzyme 1α (IRE1α/ERN1)[1–4]. Various tumors display altered activation of different UPR branches according to cell types and tissue origins; UPR activation is crucial for tumor cell adaption to ER stress[7,8]. However, the molecular mechanisms by which different UPR branches in tumors are selectively activated remain poorly understood.

IRE1α is the ancestral branch of the ER stress response and is an ER-resident transmembrane protein acting as a proximal sensor of the UPR[9]. Among various protein kinases, IRE1α has been proposed as a major contributor to tumor progression[10]. In some cancers, constitutive activation of the IRE1α pathway is linked to cell survival and tumor progression under stress[11–14]. Indeed, the IRE1α pathway is linked to the expression of several tumor-promoting factors[15–17]. IRE1α is activated by autophosphorylation and oligomerization, resulting in activation of its endoribonuclease (RNase) to cleavage and initiated the splicing of the X-box binding protein 1 (XBP1) mRNA. XBP1 is a unique transcription factor that regulates genes responsible for ER-associated degradation (ERAD) and protein folding. Aside from RNase activity of IRE1α, phosphorylation of its kinase domain recruits TRAF2 to facilitate JNK and p38 MAPK activation under ER stress[18]. Activated IRE1α also induces ERK MAPK activation through dissociation from the SH2/SH3 domain-containing adaptor Nck upon ER stress[19]. The activation of IRE1α is regulated by the UPRosome, a complex protein platform at the ER membrane[18]. Bax inhibitor-1 (BI-1) forms a complex with the cytosolic domain of IRE1α and inhibits IRE1α signaling[20]. BAX, BAK, ASK1-interacting protein (AIP1), and Hsp72 are also associated with the cytoplasmic domain of IRE1α and enhance its activation[21–23]. Thus, the IRE1α binding partners are key regulators of selective activation of its signaling pathway and determination of cell fate upon ER stress.

Protein kinase C substrate 80K-H (PRKCSH/Hepatocystin) normally resides in the ER lumen, where it functions as the noncatalytic β subunit of glucosidase II (GII)[24–26]. It forms the heterodimeric GII complex with glucosidase II α subunit (GIIα), which processes newly synthesized glycoproteins and is involved in ER protein quality control[24–26]. PRKCSH consists of multiple domains, including a signal sequence for translocation across the ER membrane, an N-terminal GIIα-binding (G2B) domain, a putative coiled-coil segment, a glutamic acid and proline-rich (E/P) segment, and a C-terminal mannose 6-phosphate receptor homology (MRH) domain followed by an HDEL signal sequence for ER retention[27,28]. ER-translocation of PRKCSH is required for the expression and retention of GIIα in the ER lumen and maintaining optimal GII activity[29]. Genetic loss of PRKCSH is involved in autosomal dominant polycystic liver disease (ADPLD)[30,31]. The underlying molecular mechanism of ADPLD is that PRKCSH deficiency impairs GII-dependent glucose trimming of PKD2 (a TPR family channel) and induces its improper folding; PKD2 is subsequently degraded by ERAD[32]. Somatic loss of PRKCSH alleles results in embryonic lethality in mouse and *Xenopus laevis* models[33,34]. Furthermore, increased PRKCSH expression is positively correlated with the progression of lymph node metastasis in breast cancer[35]. It is also associated with tumor resistance to chemotherapeutic drugs such as gefitinib, an EGFR inhibitor, through an unidentified mechanism[36]. Here, we demonstrate a role of PRKCSH in specific regulation of UPR signaling, which is potentially involved in tumorigenesis.

## Results

**PRKCSH expression is associated with tumorigenesis**. To investigate the association of PRKCSH with tumorigenesis, we analyzed the relative expression levels of PRKCSH by using the complete data sets of human tumor tissues of The Cancer Genome Atlas (TCGA). The expression of the *PRKCSH* gene was significantly upregulated in various cancer tissues such as glioblastoma multiforme, esophageal carcinoma, lymphoid neoplasm diffuse large B-cell lymphoma, thymoma, liver hepatocellular carcinoma, pancreatic adenocarcinoma, stomach adenocarcinoma, and skin cutaneous melanoma ((Fig. 1a, b; Supplementary Fig. 1a). Subsequently, we also analyzed PRKCSH expression in human tumor tissues using the data available from the National Center for Biotechnology Information (NCBI) Gene Expression Omnibus (GEO) database. The expression of the *PRKCSH* gene was significantly upregulated in liver, colon, gastric, breast, and lung cancer tissues (Supplementary Fig. 1b). Immunohistochemical (IHC) analysis of an liver cancer tissue microarray also revealed that the incidence of PRKCSH positivity was higher in tumor tissues (positive samples: 45 out of 58; 77.6%) than in nontumor tissues (positive samples: 10 out of 59; 16.9%) (Fig. 1c, d; Supplementary Fig. 1c). Immunoblot analysis also showed that the expression level of PRKCSH is increased in hepatoma cell lines (HepG2 and Huh-7) compared to that of normal liver cell lines (CCL-13 and L02) (Fig. 1e). These data indicated that an increased level of PRKCSH is implicated in tumorigenesis; hence, we further analyzed the relationship between PRKCSH expression and clinicopathological parameters by using the same IHC data set. PRKCSH expression was significantly correlated with both extrahepatic metastasis (chi-square test, $P = 0.029$) and TNM classification of malignant tumors (TNM) stage (chi-square test, $P = 0.028$) (Fig. 1f, Supplementary Table 1). Analysis of the prognostic association of patient survival with *PRKCSH* mRNA level by using the data from TCGA and the European Bioinformatics Institute of the European Molecular Biology Laboratories data (EMBL-EBI) revealed that patients with high expression showed poor survival rate (Fig. 1g; Supplementary Fig. 1d). These results suggest that a potential function of PRKCSH is closely related to HCC tumorigenesis and progression.

**PRKCSH regulates the IRE1α-XBP1 and -MAPK pathways**. To define the possible function of PRKCSH in the regulation of UPR, we investigated the involvement of PRKCSH in the IRE1α–XBP1 pathway by using PRKCSH overexpression (L02-PRK) and knockout (L02-PRK KO) L02 normal liver cells. The ER localization of ectopically expressed PRKCSH was confirmed by immunocytochemistry[37,38] (Supplementary Fig. 2a). Upon treatment with tunicamycin (TM), a typical ER stress inducer, the level of spliced *XBP1* mRNA was increased in L02-PRK cells compared to control cells (L02-Mock), whereas the levels of total *XBP1* mRNA were similar (Fig. 2a, Supplementary Fig. 3a, b). Similar results were obtained for the level of spliced XBP1 protein (Fig. 2b). Also, the level of spliced XBP1 protein was increased in a dose dependent manner by TM treatment; this increase was stronger in L02-PRK cells than in control cells (Fig. 2c). Upon glucose deprivation, the level of spliced XBP1 protein was also

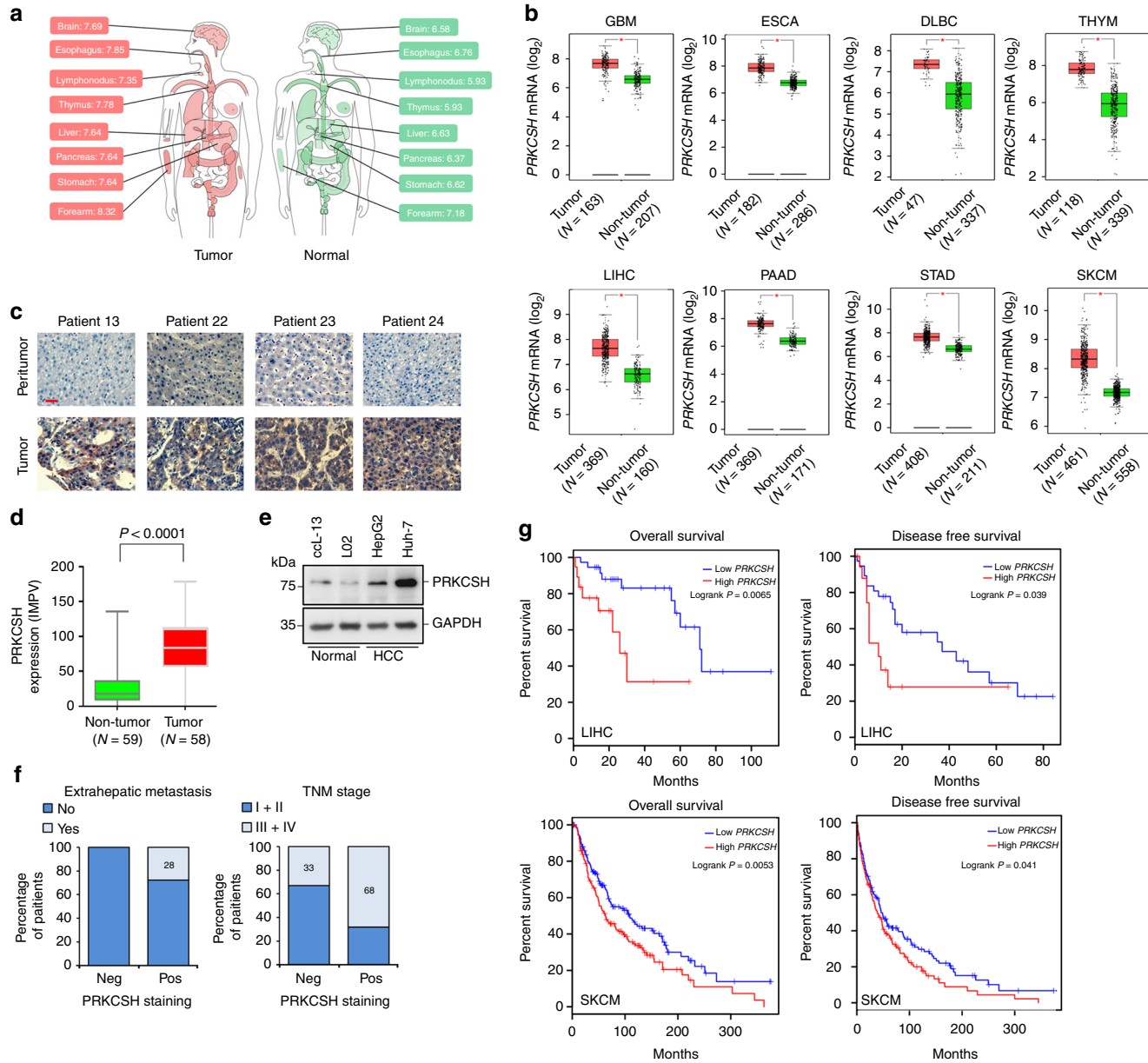

**Fig. 1** The level of PRKCSH is increased in various cancer tissues. **a** Map of human organs in which the expression of *PRKCSH* mRNA is significantly higher in tumors than in nontumor. Gene-expression datasets from TCGA were analyzed by using the GEPIA web tool. **b** Scatter plots showing relative levels of *PRKCSH* mRNA in nontumor and tumor tissues. Median expression levels in each group are indicated by horizontal lines and these values are shown in (**a**). One-way ANOVA; *$P < 0.01$. **c** Representative immunohistochemical images, and **d** relative levels of PRKCSH protein. The intensity of PRKCSH staining in paraffin-embedded sections of liver tissue arrays (nontumor, $n = 59$; tumor, $n = 58$) was calculated as described in Supplementary Fig. 1c. Scale bars represent 20 μm. The center line of the boxplots denotes the median, the bounds of the box indicate 25–75% and the whiskers represent 5–95%, respectively. Significance of the differences between the two categories was determined by the Student *t* test ($P < 0.0001$). **e** Immunoblot analysis of PRKCSH expression in human liver cell lines. **f** Relationship between PRKCSH expression and clinicopathological features. Among the clinicopathological data of the liver tissue array, extrahepatic metastasis and TNM stage were positively related to PRKCSH staining (Supplementary Table 1). **g** Overall survival and disease-free survival curves for patient groups with high and low *PRKCSH* expression level. The data were obtained from TCGA datasets. Significance of the differences between the two categories was determined by Log-rank test

higher in L02-PRK cells than in L02-Mock cells (Fig. 2d). Conversely, the levels of spliced *XBP1* mRNA and its protein were attenuated in L02-PRK KO cells under ER stress (Fig. 2e, f). Subsequently, we assessed the role of PRKCSH in the ER stress-mediated activation of MAPKs. Activation of ERK1/2 and JNK1/2 was increased in L02-PRK cells compared to L02-Mock cells (Fig. 2g), but was attenuated in L02-PRK KO cells upon TM treatment (Fig. 2h) or glucose deprivation (Fig. 2i). Finally, we examined the effect of PRKCSH on the expression of the XBP1

target genes *ERDJ4*, *GRP78*, *Sec61A1*, and *p58IPK* (Supplementary Fig. 3a)[39,40]. Upon TM treatment, the expression of these genes was significantly higher in L02-PRK cells than in L02-Mock cells, but was lower in L02-PRK KO (Fig. 2j). Overall, these results indicate that PRKCSH boosts the activation of both IRE1α–XBP1 and IRE1α–MAPK pathways under ER stress.

**PRKCSH contributes to selective activation of IRE1α pathway.** Some investigators have reported that PRKCSH is localized in the

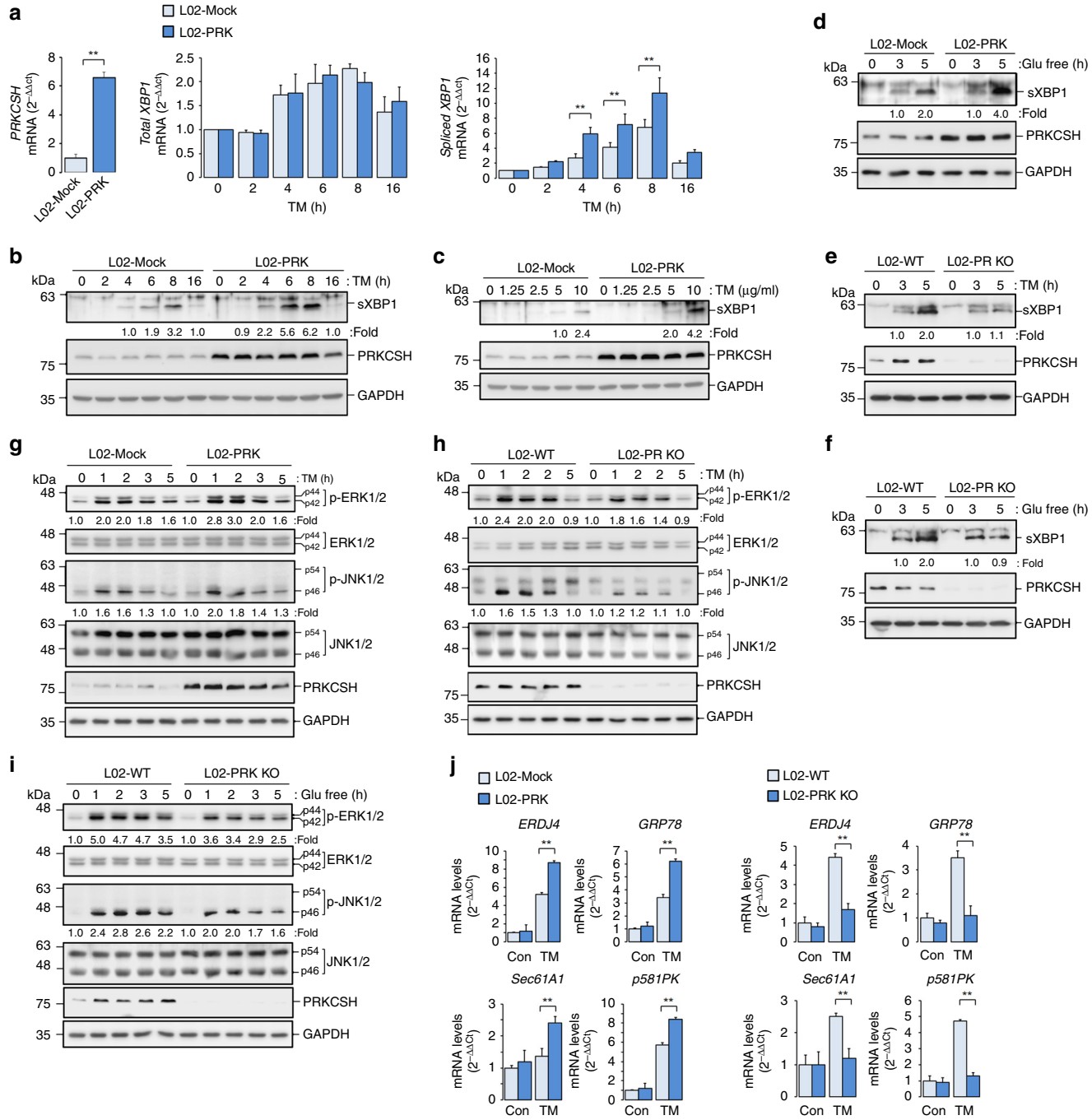

**Fig. 2** PRKCSH promotes activation of the IRE1α pathway under ER stress. **a** Quantitative real-time PCR analysis of total *XBP1* and spliced *XBP1* mRNA levels in L02-Mock and L02-PRK cells treated with 10 μg/mL TM for the indicated time. **b, c** Immunoblot analysis of spliced XBP1 protein (sXBP1) levels in L02-Mock and L02-PRK cells treated with 10 μg/mL TM for the indicated time (**b**) or the indicated doses of TM for 5 h (**c**). **d** Immunoblot analysis of sXBP1 levels in L02-Mock and L02-PRK cells treated with glucose-free medium for the indicated time. **e, f** Immunoblot analysis of sXBP1 levels in L02-WT and L02-PRK KO cells treated with 10 μg/mL TM (**e**) or glucose-free medium (**f**) for the indicated time. **g** Immunoblot analysis of ERK and JNK phosphorylation in L02-Mock and L02-PRK cells treated with 10 μg/mL TM for the indicated time. **h, i** Immunoblot analysis of ERK and JNK phosphorylation in L02-WT and L02-PRK KO cells treated with 10 μg/mL TM (**h**) or glucose-free medium (**i**) for the indicated time. **j** Quantitative real-time PCR analysis of UPR genes in L02-Mock and L02-PRK cells, or L02-WT and L02-PRK KO cells treated with 10 μg/mL TM for 5 h. sXBP1, phosphorylated ERK, and phosphorylated JNK levels shown below the blots were normalized to GAPDH, total ERKs, and total JNKs, respectively. For quantitative real-time PCR analysis, Data are represented as mean ± SEM of four samples. One-way ANOVA; $^{**}P < 0.01$

nucleus in breast cancer models[41]. However, we did not observe nuclear localization of PRKCSH in liver tumor tissues (Supplementary Fig. 1c) or hepatoma cells (Supplementary Fig. 2b, c). Analysis of co-localization of PRKCSH and calnexin showed that PRKCSH is primarily localized in the ER in hepatoma cells

(Supplementary Fig. 2b), indicating that the PRKCSH level is increased in the ER of these cells in comparison with normal liver cells. To investigate the potential role of PRKCSH in the regulation of the ER stress response in HCC cell models, we compared the levels of spliced XBP1 and activated MAPKs between

PRKCSH-silenced and control hepatoma cells. The splicing of *XBP1* mRNA was significantly downregulated in PRKCSH-knockdown HepG2 cells in comparison with control cells upon TM treatment, whereas the expression of total *XBP1* mRNA was not different between these cells (Fig. 3a). Comparable results were also observed in Huh-7 hepatoma cells (Fig. 3b, Supplementary Fig. 3c). Silencing PRKCSH in both hepatoma cell lines decreased the level of spliced XBP1 protein (Fig. 3c, d). To further confirm the potential role of PRKCSH in XBP1 splicing under ER stress, we used Huh-PRK KO cells. The level of spliced XBP1 protein was also decreased in these cells upon TM treatment (Fig. 3e). Consistent with the effect of TM on XBP1 splicing, activation of ERK1/2, and JNK1/2 MAPKs upon TM treatment was decreased by PRKCSH silencing in HepG2 cells (Fig. 3f). Furthermore, the expression levels of IRE1α–XBP1 target genes were also decreased by PRKCSH silencing in Huh-7 hepatoma cells (Fig. 3g). PRKCSH knockout in Huh-7 cells also markedly reduced the expression levels these genes (Fig. 3g). Lastly, since we showed that the levels of *PRKCSH* mRNA were elevated in several tumor tissues (Fig. 1b), we checked whether the expression levels of XBP1 target genes were elevated in human tumors. We found that all the tested XBP1 target genes were upregulated in tumor tissues compared to nontumor tissues (Fig. 3h, Supplementary Fig. 4). The expression level of each of these genes was positively correlated with the level of *PRKCSH* mRNA (Fig. 3h, Supplementary Fig. 4). These results suggest that the upregulated PRKCSH boosts the activation of the IRE1α pathway during tumorigenesis.

We then examined the effect of PRKCSH on the activation of the other UPR branches, the PERK and ATF6 pathways. PRKCSH-overexpression, -silencing, or knockout had little effect on PERK phosphorylation or ATF4 expression (Supplementary Fig. 5a–d). In agreement with this result, there were no significant changes in the expression level of *ERO1LB*, a target gene of the PERK pathway, in L02-PRK, L02-PRK KO, PRKCSH-silenced, and Huh-PRK KO cells in comparison with control cells (Supplementary Fig. 5e). Furthermore, silencing or overexpression of PRKCSH resulted in no meaningful difference in the levels of ATF6 activation (Supplementary Fig. 5g, h) or the expression of its target genes, as indicated by the total level of *XBP1* mRNA (Figs 2a, 3a, b). Taken together, our results indicate that PRKCSH has a specific function in the regulation of the IRE1α pathway in the ER stress response.

**PRKCSH promotes phosphorylation and oligomerization of IRE1α.** Under ER stress, IRE1α can be phosphorylated in the linker region, kinase activation domain, and the RNase domain[42]. Phosphorylation of the kinase domain activates the downstream MAPKs and RNase domain of IRE1α[42]. Thus, to define the molecular mechanism by which PRKCSH activates the IRE1α pathway, we determined the phosphorylation of IRE1α. Upon TM treatment, the level of IREα phosphorylation was higher in L02-PRK cells than in L02-Mock cells (Fig. 4a). In contrast, L02-PRK KO cells showed a decreased level of IRE1α phosphorylation compared to wild-type (WT) cells after either TM treatment (Fig. 4b) or glucose starvation (Fig. 4c). PRKCSH-silenced hepatoma cells and Huh-PRK KO cells showed a remarkable reduction in IRE1α phosphorylation upon TM treatment (Fig. 4d–f). Since IRE1α oligomerization is a key step for UPRosome formation and triggering the subsequent XBP1 splicing[43–45], we then investigated the effect of PRKCSH on IRE1α oligomerization under ER stress. The oligomerization of IRE1α was monitored as the formation of foci in cells transfected with a Flag and Venus-tagged fluorescent human IRE1α fusion construct (IRE1α-FV)[46]. To examine whether ectopic overexpression of IRE1α results in formation of inclusion bodies containing proteasome components, we performed immunocytochemical analysis. The IRE1α-FV protein was not associated with proteasome under either resting or ER stress condition, suggesting that the fluorescent foci were derived from oligomerized IRE1α in the ER (Supplementary Fig. 6). Upon TM treatment, IRE1α oligomerization was higher in L02-PRK cells (up to 64% foci-positive cells) than in L02-Mock cells (up to 42% foci-positive cells) (Fig. 4g). In Huh-7 hepatoma cells, the oligomerization of IRE1α was drastically increased upon TM treatment (up to 65% foci-positive cells), whereas silencing PRKCSH in this cell line considerably reduced it under ER stress (up to 30% foci-positive cells) (Fig. 4h). These results indicate that PRKCSH promotes activation of IRE1α under ER stress.

Finally, we investigated whether IRE1α is necessary for PRKCSH-mediated XBP1 splicing and MAPK activation. Silencing of IRE1α in L02-PRK cells markedly reduced XBP1 splicing and ERK1/2 MAPK activation (Fig. 4i–k), suggesting that IRE1α is required for PRKCSH-mediated XBP1 splicing and MAPK activation. To confirm this result, we investigated the impact of IRE1α overexpression on XBP1 splicing and MAPK activation in PRKCSH-deficient cells. Overexpression of IRE1α increased XBP1 splicing and MAPK activation in L02-PRK KO cells (Fig. 4i–k). Taken together, our results indicate that IRE1α is a downstream target of PRKCSH during ER stress response.

**PRKCSH specifically interacts with IRE1α under ER stress.** To determine how PRKCSH promotes the activation of IRE1α under ER stress, we investigated whether the two proteins physically interact under ER stress by using co-immunoprecipitation (co-IP) from lysates of L02 normal liver cells. Association between PRKCSH and IRE1α was not observed under resting conditions; however, strong association was observed at 60 min of TM treatment, concomitant with the dissociation of GRP78 from IRE1α (Fig. 5a). Interestingly, this association was disappeared at 120 min of TM treatment, although dissociation of GRP78 from IRE1α persisted, consistent with previously reported observations[18]. Similar results were obtained under glucose deprivation (Fig. 5a). TM treatment also increased the association between endogenous PRKCSH and IRE1α in Huh-7 hepatoma cells (Fig. 5a). Interestingly, the interaction was more sustained than that in L02 cells. To confirm the interaction of PRKCSH with IRE1α, we performed co-IP using lysates of L02 cells transfected with a Flag-tagged PRKCSH construct (PRK-Flag). Interaction between PRK-Flag and endogenous IRE1α was increased upon TM treatment (Fig. 5b). We further investigated this interaction using lysates of L02 cells transfected with IRE1α-FV. Association of endogenous PRKCSH with IRE1α-FV was marginal in resting cells but was increased upon TM treatment (Fig. 5c). These results indicated that ER stress induces a physical association between PRKCSH and IRE1α. We further determined whether PRKCSH interacts with PERK under TM treatment. PRKCSH did not interact with PERK under either resting or ER stress conditions (Supplementary Fig. 5f), although PERK was activated under the same ER stress conditions (Supplementary Fig. 5a, b). Overall, our data demonstrate that PRKCSH specifically interacts with IRE1α under ER stress.

**PRKCSH E/P domain is essential for boosting IRE1α activation.** To determine whether the interaction between PRKCSH and IRE1α is direct, we performed an in vitro pull-down assay with GST-PRKCSH deletion mutants and purified IRE1α. IRE1α interacted with the C-terminal domain of PRKCSH, but not with its N-terminal domain (Fig. 5d), indicating that PRKCSH directly interacts with IRE1α. To further determine the region of

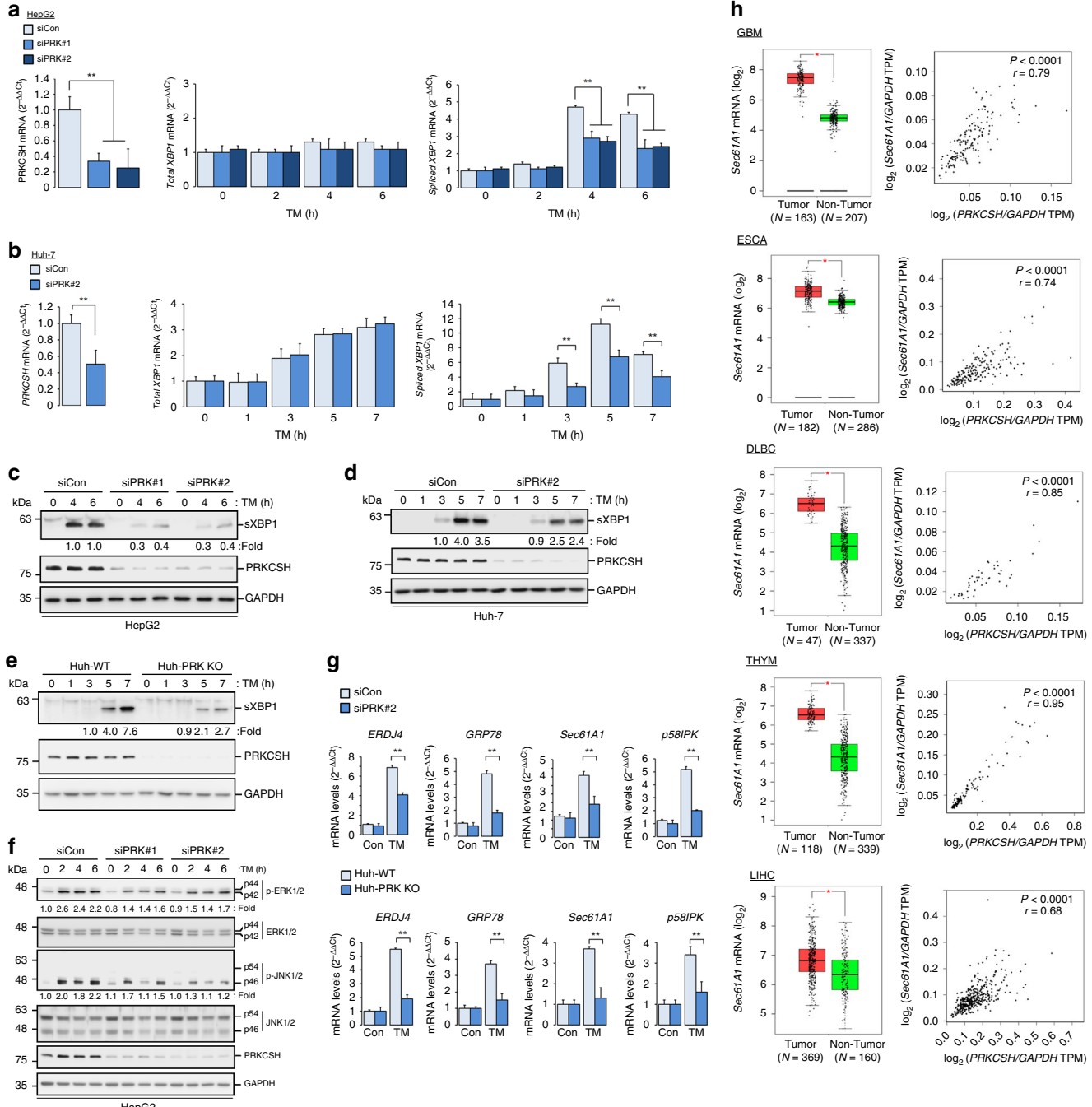

**Fig. 3** PRKCSH is required for boosting activation of the IRE1α pathway in hepatoma cells under ER stress. **a**, **b** Quantitative real-time PCR analysis of total *XBP1* and spliced *XBP1* expression in PRKCSH-silenced HepG2 (**a**) and Huh-7 (**b**) hepatoma cells treated with 10 μg/mL TM for the indicated time. **c**, **d** Immunoblot analysis of sXBP1 protein levels in PRKCSH-silenced HepG2 (**c**) and Huh-7 (**d**) cells treated with 10 μg/mL TM for the indicated time. **e** Immunoblot analysis of sXBP1 protein levels in Huh-WT and Huh-PRK KO cells treated with 10 μg/mL TM for the indicated time. **f** Immunoblot analysis of ERK and JNK phosphorylation in PRKCSH-silenced HepG2 cells treated with 10 μg/mL TM for the indicated time. **g** Quantitative real-time PCR analysis of UPR genes in PRKCSH-silenced Huh-7 cells or Huh-PRK KO cells treated with 10 μg/mL TM for 5 h. sXBP1, phosphorylated ERK, and phosphorylated JNK levels shown below the blots were normalized to GAPDH, total ERKs, and total JNKs, respectively. For quantitative real-time PCR analysis, Data are represented as mean ± SEM from three independent experiments. One-way ANOVA; **$P < 0.01$. **h** Expression of *Sec61A1* as representative XBP1 target gene and its correlation with the levels of *PRKCSH* mRNA in glioblastoma multiforme (GBM), esophageal carcinoma (ESCA), lymphoid neoplasm diffuse large B-cell lymphoma (DLBC), thymoma (THYM), and liver hepatocellular carcinoma (LIHC) tissues. TCGA data were analyzed by using the GEPIA web tool. Scatter plots show relative expression of *Sec61A1* mRNA in non-tumor and tumor tissues. The median expression levels in each group are indicated by horizontal lines. Significance of the differences between nontumor and tumor (left) was determined by one-way ANOVA ($P < 0.001$). Correlation between Sec61A1 and PRKCSH (right) was determined by Pearson's rank correlation coefficient. *r* Pearson correlation coefficient, *P* Pearson *p* value

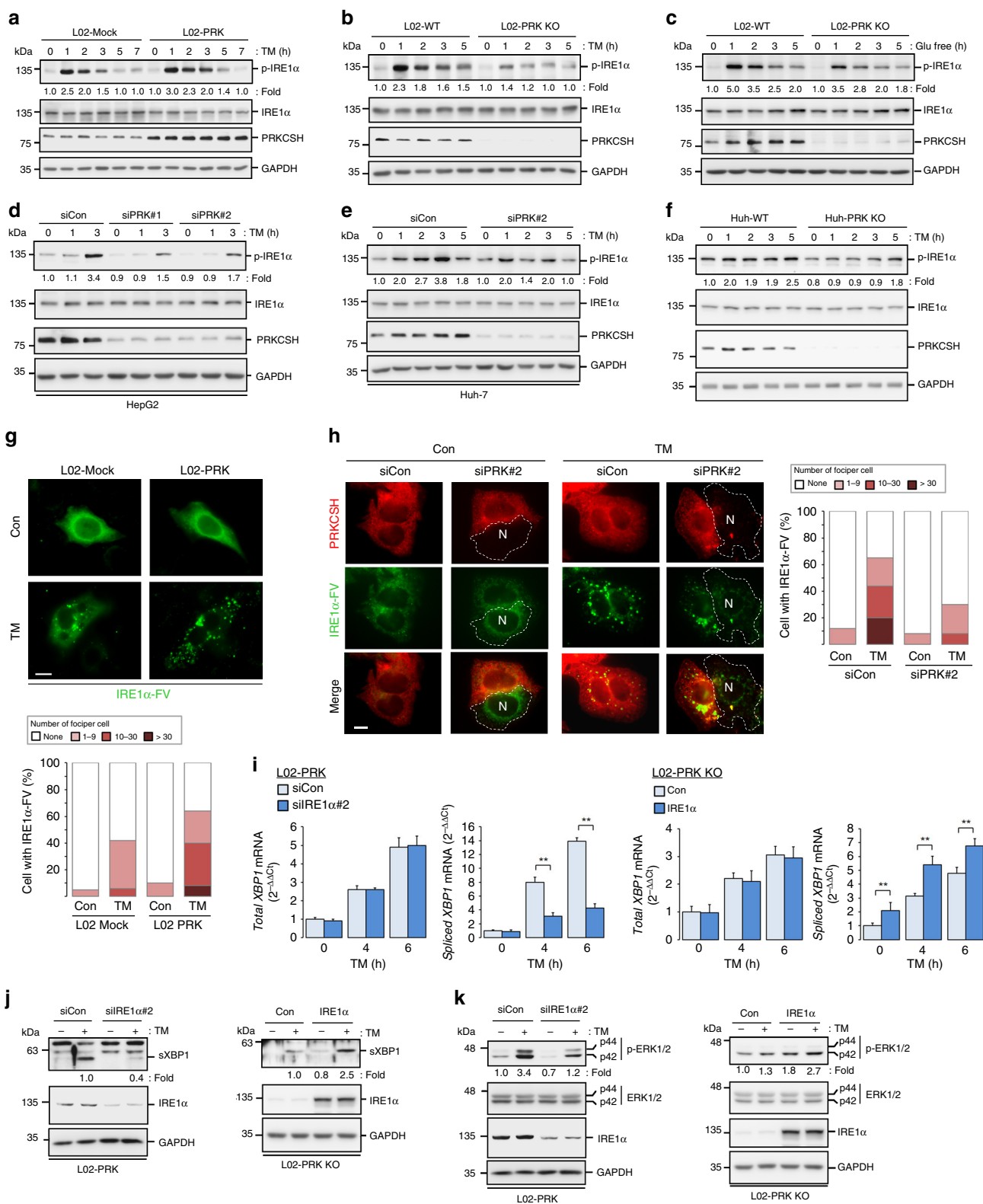

PRKCSH responsible for association with IRE1α, we performed co-IP from lysates cells transfected with Flag-tagged PRKCSH mutants. We first confirmed that all these mutant proteins except ΔS/G2B are localized in the ER, similar to WT (Fig. 5f, Supplementary Fig. 7a). Co-IP assay revealed that the internal E/P domain of PRKCSH is required for the interaction with IRE1α upon ER stress (Fig. 5e, f, Supplementary Fig. 7b). Unlike the

ΔG2B mutant, the ΔS/G2B mutant, which lack the N-terminal ER localization signal, did not interact with IRE1α (Fig. 5f). This result indicates that PRKCSH interacts with IRE1α in the ER but not in the cytoplasm. Interestingly, unlike WT PRKCSH, mutants lacking the G2B region (ΔG2B and E/P) interacted with IRE1α even under resting conditions (Fig. 5e, f, Supplementary Fig. 7b). The inhibitory effect of the G2B domain on PRKCSH binding to

**Fig. 4** PRKCSH promotes phosphorylation and oligomerization of IRE1α under ER stress. **a** Immunoblot analysis of IRE1α phosphorylation in L02-Mock and L02-PRK cells treated with 10 μg/mL TM for the indicated time. **b**, **c** Immunoblot analysis of IRE1α phosphorylation in L02-WT and L02-PRK KO cells treated with 10 μg/mL TM (**b**) or glucose-free medium (**c**) for the indicated time. **d**, **e** Immunoblot analysis of IRE1α phosphorylation in PRKCSH-silenced HepG2 (**d**) and Huh-7 (**e**) hepatoma cells treated with 10 μg/mL TM for the indicated time. **f** Immunoblot analysis of IRE1α phosphorylation in Huh-WT and Huh-PRK KO cells treated with 10 μg/mL TM for the indicated time. Phosphorylated IRE1α levels shown below the blots were normalized to total IRE1α. **g** IRE1α foci in L02-Mock and L02-PRK cells transfected with IRE1α-FV construct and treated with 10 μg/mL TM for 4 h. The percentage of cells with IRE1α foci per total cells with fluorescent IRE1α is shown below the images. The number of IRE1α foci per cell was counted for at least 100 cells with fluorescent IREα-FV. Scale bars represent 5 μm. **h** IRE1α foci in PRKCSH-silenced Huh-7 cells transfected with IRE1α-FV construct and treated with 10 μg/mL TM for 4 h. PRKCSH is counterstained and its knockdown cells are indicated as dotted line. N indicates the nucleus. Scale bars represent 5 μm. The percentage of cells with IRE1α foci per total cells with fluorescent IRE1α is shown on the right. The number of IRE1α foci per cell was counted for at least 100 cells with fluorescent IREα-FV. **i** Quantitative real-time PCR analysis of total XBP1 and spliced XBP1 expression in IRE1α-knockdown L02-PRK cells andIRE1α-overexpressing L02-PRK KO cells. **j** Immunoblot analysis of sXBP1 protein levels in these cells. **k** Immunoblot analysis of ERK phosphorylation. The numbers indicate the relative levels of sXBP1 and phosphorylated ERKs normalized to the levels of GAPDH or total ERKs, respectively. Quantitative real-time PCR data are shown as mean ± SEM of four independent experiments. One-way ANOVA; **$P < 0.01$

IRE1α suggests that it regulates the specific binding of PRKCSH to IRE1α only when ER stress is present. Because the interaction between the GIIα subunit and PRKCSH is necessary for GII activity[24–29], we examined the potential impact of ER stress on GII activity. Co-IP assay revealed that ER stress induced the dissociation of the GIIα–PRKCSH complex in both L02 and Huh-7 cells (Fig. 5g). This result suggest that GII activity is reduced during ER stress response.

To investigate the impact of interaction between IRE1α and PRKCSH on IRE1α activation, we evaluated the levels of IRE1α phosphorylation, XBP1 splicing, and ERK activation. Expression of WT PRKCSH enhanced the ER stress-induced phosphorylation of IRE1α, whereas its level in cells expressing the ΔE/P mutant was similar to that of control cells (Fig. 6a). Consistent with this result, the levels of spliced XBP1 protein and ERK activation were also increased in cells transfected with WT PRKCSH but not with the ΔE/P mutant (Fig. 6b, c). Expression of the ΔG2B or E/P mutant increased IRE1α phosphorylation and XBP1 splicing upon TM treatment, whereas expression of the ΔS/G2B mutant showed little effect in comparison with control cells (Fig. 6d–g). Both the ΔG2B and E/P mutants interacted with but not activate IRE1α under resting conditions (Fig. 5f). These results indicate that PRKCSH–IRE1α interaction as such does not trigger IRE1α activation but boosts it upon ER stress.

Taken together, our results suggest that PRKCSH is involved in regulation of GII activity through interaction with the GIIα subunit under resting condition; however, it regulates IRE1α activation through the E/P domain-mediated interaction with IRE1α under ER stress (Fig. 5h).

**PRKCSH is crucial for tumor resistance to ER stress.** The IRE1 signaling branch is responsible for tumor resistance against ER stress-induced cell death and for the regulation of the expression of tumor-promoting factors[13,16,17]. To investigate the effect of PRKCSH on ER stress-induced cell death, we monitored apoptotic cell death and PARP1 cleavage in PRKCSH-overexpressing and -silenced cells. Importantly, overexpression of PRKCSH rendered L02 cells strongly resistant to ER stress-induced cell death upon treatment with TM or another ER stressor, thapsigargin (TG), and reduced PARP1 cleavage (Fig. 7a, Supplementary Fig. 8a). PRKCSH silencing in Huh-7 cells significantly sensitized them to ER stress-induced cell death and increased PARP1 cleavage upon TM or TG treatment (Fig. 7b, c, Supplementary Fig. 8b, c). Similar results were also obtained in Huh-PRK KO cells upon TM treatment (Fig. 7d, Supplementary Fig. 8d). To examine the effect of PRKCSH on long-term overall survival upon ER stress, we performed cell survival assay using TM-treated L02-PRK and Huh-PRK KO cells. Overexpression of PRKCSH increased survival of L02 cells under ER stress.

Conversely, knockout of PRKCSH reduced survival of Huh-7 cells under ER stress (Fig. 7e). In addition, we determined whether PRKCSH contributes to cytoprotection against other stress condition such as nutrient starvation. Similar to the results for TM treatment, overexpression of PRKCSH also increased survival of L02 cells upon nutrient starvation, whereas its knockout reduced survival of Huh-7 cells (Fig. 7e). To evaluate the in vivo relevance of our study, we performed in vivo tumor growth experiment in xenograft nude mouse model using Huh-PRK KO and WT cells. Loss of PRKCSH reduced in vivo tumor growth under physiological stress condition (Fig. 7f, Supplementary Fig. 9). To investigate whether the E/P domain is required for the cytoprotective effect of PRKCSH against ER stress, we monitored ER stress-induced apoptotic cell death using cells overexpressing WT or mutant PRKCSH (ΔE/P and MRH). Deletion of the E/P domain resulted in a loss of cytoprotective function of PRKCSH (Fig. 7g, Supplementary Fig. 8e). We investigated whether IRE1α is required for PRKCSH-mediated cell survival under ER stress. IRE1α silencing attenuated the resistance of L02-PRK cells to ER stress-induced cell death (Fig. 7h, Supplementary Fig. 8f). This effect was confirmed by an increase in ER stress-induced PARP1 cleavage in IRE1α-knockdown cells (Fig. 7h). Next, we investigated the effect of PRKCSH on the regulation of ER stress-induced expression of tumor-promoting factors (Supplementary Fig. 3d). Overexpression of PRKCSH increased the levels of TNFA, IL8, and VEGF mRNAs upon TM treatment (Fig. 8a). Conversely, PRKCSH silencing in HepG2 and Huh-7 hepatoma cells significantly attenuated TM-induced expression of these genes (Fig. 8b, c). Finally, we investigated whether IRE1α is required for PRKCSH-mediated expression of tumor-promoting cytokines under ER stress. The increased expression of tumor-promoting factors in PRK cells was reduced by IRE1α silencing (Fig. 8d). Overall, these results suggest that the PRKCSH–IRE1α signaling axis is crucial for adaptation of tumor cells to ER stress.

## Discussion

Previous studies have shown that PRKCSH is the β subunit of GII; PRKCSH determines the fate of glycoproteins and is needed for optimal quality control of glycoproteins in the ER[24–26]. In the present study, we demonstrate a function of PRKCSH as a specific regulator of IRE1α signaling. The function is derived from the physical interaction between PRKCSH and IRE1α under ER stress. Our data provide strong evidence that PRKCSH can be a useful tumor marker in various cancer tissues and protects tumor cells from ER stress by promoting IRE1α activity (Fig. 9).

In the ER, PRKCSH is essential for GII activity, which ensures secretion properly folded glycoproteins and targeting the improperly folded ones for degradation by the ERAD pathway[24–26]. Thus, we expected that alteration of PRKCSH expression would

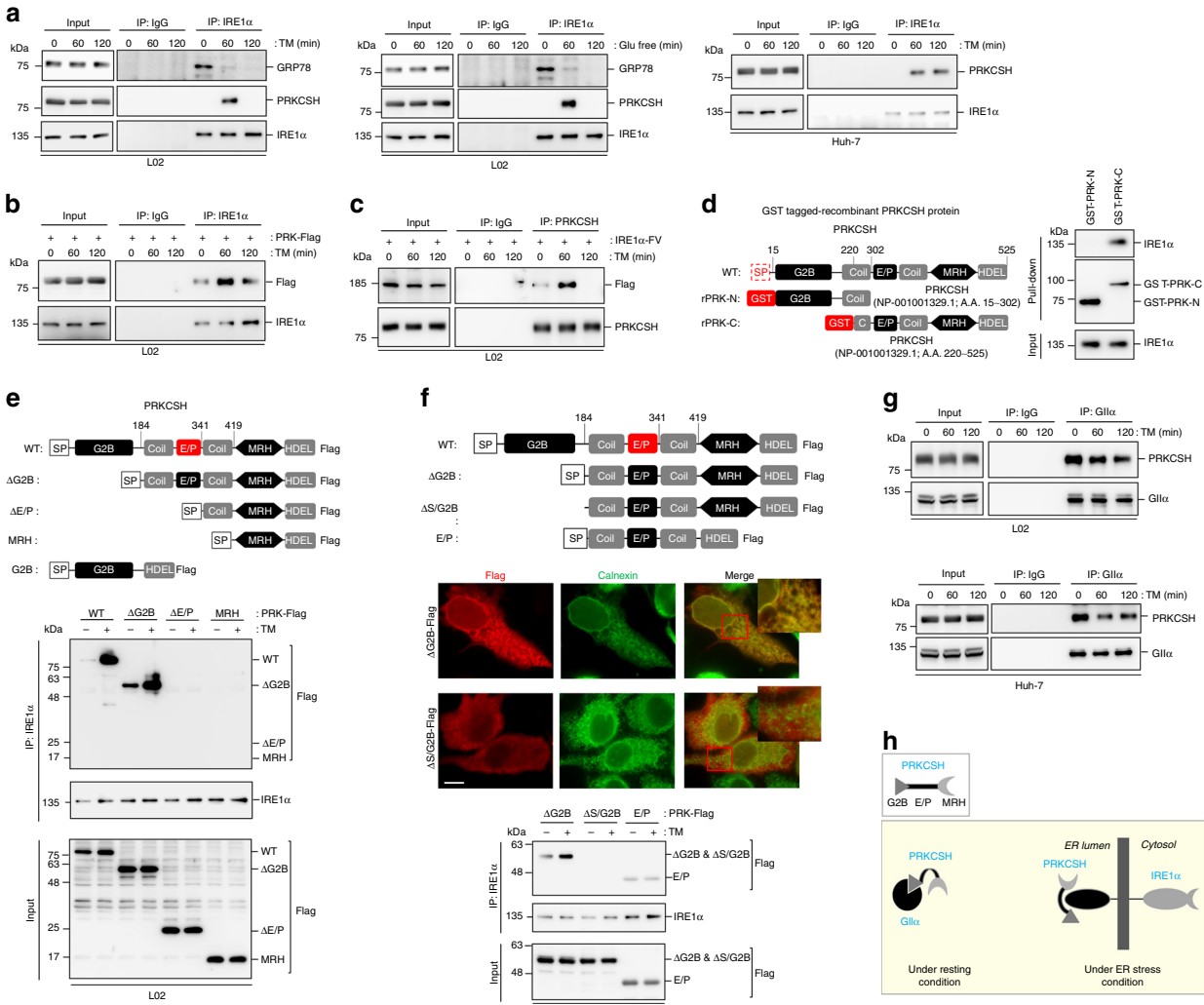

**Fig. 5** Internal E/P domain of PRKCSH is required for its interaction with IRE1α and boosting IRE1α activation under ER stress. **a** Immunoblot analysis of association between endogenous PRKCSH and IRE1α using immunoprecipitates from L02 cells treated with 10 μg/mL TM (left) or glucose-free medium (middle) for the indicated time, and from Huh-7 cells treated with 10 μg/mL TM (right) for the indicated time. **b** Immunoblot analysis of complex formation between endogenous IRE1α and Flag-tagged PRKCSH immunoprecipitated from transfected L02 cells treated with 10 μg/mL TM for the indicated time. Immunoprecipitation was performed by using anti-IRE1α antibody and normal rabbit IgG as a control antibody. **c** Immunoblot analysis of complex formation between endogenous PRKCSH and Flag-tagged IRE1α. Immunoprecipitates were prepared from transfected L02 cells treated with 10 μg/mL TM for the indicated time. Immunoprecipitation was performed by using anti-PRKCSH antibody and normal rabbit IgG. **d** Schematic diagram of the functional domains of PRKCSH and its recombinant variants (left part). Immunoblot analysis of in vitro complex formation between IRE1α and GST-tagged PRKCSH protein (right part). **e**, **f** Schematic presentation of wild-type and deletion mutants of PRKCSH (upper part). Immunoblot analysis of complex formation between endogenous IRE1α and Flag-tagged mutant PRKCSHs (lower part). Immunoprecipitates were prepared from transfected L02 cells treated with 10 μg/mL TM for 1 h. Immunocytochemical analysis of L02 cells transfected with Flag-tagged ΔG2B and ΔS/G2B mutant PRKCSH (**f**, middle). Calnexin was used as an ER marker. Scale bars represent 5 μm. **g** Immunoblot analysis of association between endogenous PRKCSH and GIIα. Immunoprecipitates were prepared from L02 (top) or Huh-7 cells (bottom) treated with 10 μg/mL TM for the indicated time. Immunoprecipitation was performed by using anti-GIIα antibody and normal rabbit IgG. **h** Model of PRKCSH complex formation. Under resting conditions, PRKCSH associates with the GIIα subunit via the G2B domain; this domain inhibits PRKCSH interaction with IRE1α. Under ER stress, PRKCSH dissociates from GIIα and then binds to IRE1α via the E/P domain

affect the activity of all three UPR branches under ER stress. However, PRKCSH had no effect on the activation of PERK and ATF6 pathways, but only promotes phosphorylation of IRE1α, leading to its oligomerization followed by XBP1 splicing and MAPK activation. These data suggest that PRKCSH functions as specific and selective regulator of the IRE1α pathway but not the other two UPR branches.

In resting conditions, GRP78 interacts with the ER–luminal domains of IRE1α, PERK, and ATF6, and prevents their activation[18]. Under ER stress, GRP78 dissociates from these sensors, thereby leading to interaction between the ER luminal domain of

each sensor and unfolded proteins, which promotes the autophosphorylation and oligomerization of IRE1α and PERK, or export of ATF6 from the ER to Golgi[18]. Since the IRE1α and PERK sensors share functionally similar luminal sensing domains[47], it has been assumed that ER stress may activate these sensors nonselectively. However, it turned out that they are selectively activated by ER stress according to cell types and ER stress inducers[7,18,48]. The selective activation of each UPR sensor may be explained by complex formation with different adaptors and modulators[18,20–23,49,50]. BAX/BAK, Bax inhibitor-1, RACK1, HSP72, and AIP1 have so far been identified as IRE1α

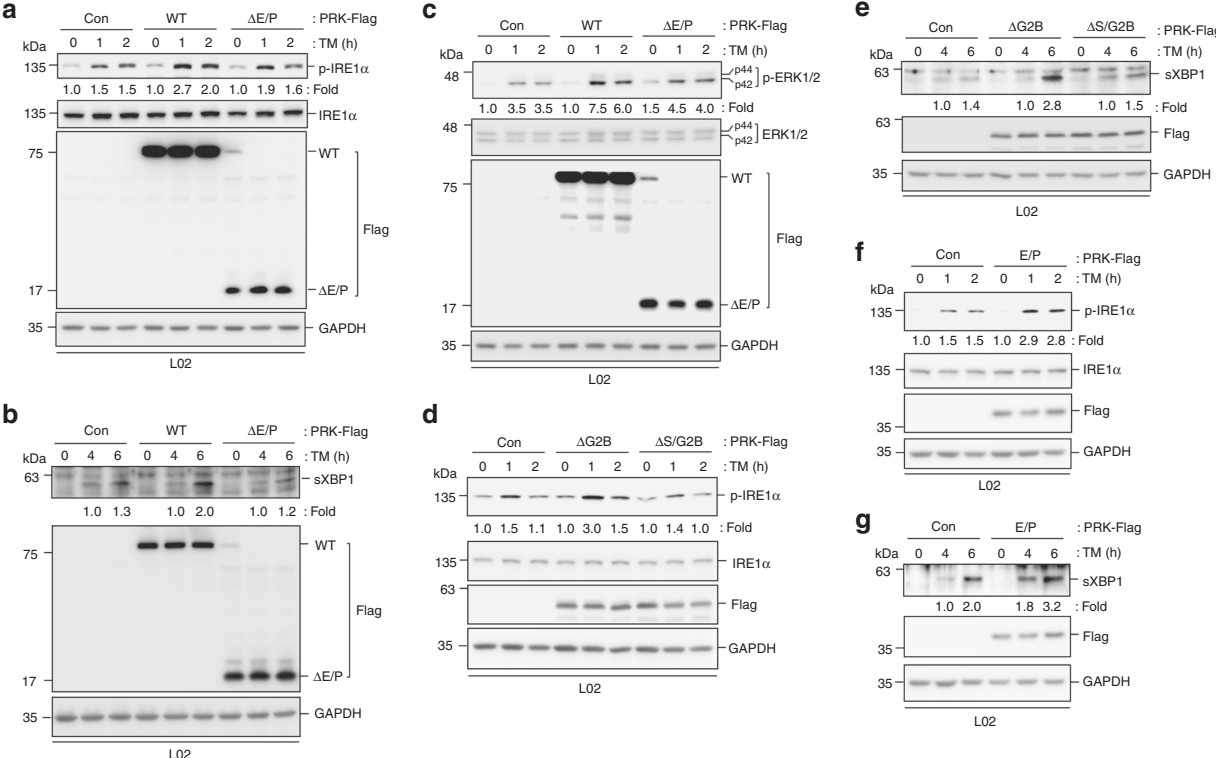

**Fig. 6** Internal E/P domain of PRKCSH is required for boosting IRE1α activation under ER stress. **a–c** Immunoblot analysis of IRE1α phosphorylation (**a**), sXBP1 protein levels (**b**), and ERK phosphorylation (**c**) from L02 cells transfected with the control vector, WT PRKCSH, or its ΔE/P mutant; cells were treated with 10 μg/mL TM for the indicated time. **d**, **e** Immunoblot analysis of IRE1α phosphorylation (**d**) and sXBP1 protein levels (**e**) in L02 cells transfected with the control, ΔG2B, or ΔS/G2B mutant plasmids; cells were treated with 10 μg/mL TM for the indicated time. **f**, **g** Immunoblot analysis of IRE1α phosphorylation (**f**) and sXBP1 protein levels (**g**) in L02 cells transfected with the control, or E/P mutant plasmids; cells were treated with 10 μg/mL TM for the indicated time

regulators[20–23]. In the present study, we showed that PRKCSH interacts only with IRE1α but not with PERK upon ER stress. Because the luminal domain of ATF6 has no structural similarity with that of IRE1α, PRKCSH is likely to be a specific binding partner of IRE1α. Our findings may explain how PRKCSH selectively boosts activation of the IRE1α pathway under ER stress.

Two domains of PRKCSH contribute to its glucosidase activity: G2B, the N-terminal GIIα-binding domain, and MRH, the C-terminal N-glycan mannose recognition domain[27,28]. Binding of the G2B domain of PRKCSH to the GIIα subunit is required for maintaining the GIIα level in the ER and GII enzymatic activity[29]. The proline-rich E/P segment of PRKCSH is positioned between the G2B and MRH domains. Several studies have proposed that proline-repeat segments merely act as spacers, or are critical for the activity of proteins and formation of complexes between them[51–53]. However, a possible function of this segment in PRKCSH has not been elucidated. In the current study, we demonstrate that the E/P domain of PRKCSH is essential for direct interaction with IRE1α. Our results obtained using various deletion mutants of PRKCSH demonstrate that the interaction between PRKCSH and IRE1α is required for boosting the activation of IRE1α upon ER stress, but is not sufficient for triggering its activation. Interestingly, deletion of the G2B domain results in PRKCSH interaction with IRE1α even under resting conditions, not only under ER stress. These data suggest that under resting conditions, the interaction of the G2B domain with the GIIα subunit might competitively inhibit PRKCSH-mediated promoting IRE1α activation. Interaction between PRKCSH and IRE1α is accompanied by dissociation of PRKCSH from the GIIα subunit

under ER stress. Thus, the inhibition of IRE1α binding by the G2B domain may allow the boosting effect on IRE1α activation specifically during ER stress. Therefore, we suggest that PRKCSH contributes to controlling ER protein quality through regulation of IRE1α activity under ER stress, although ER stress induces the dissociation of the GIIα subunit complex with PRKCSH, which may reduce GII activity.

Under ER stress, the kinase domain of IRE1α is autophosphorylated, which activates the downstream MAPKs and the RNase domain of IRE1α[42]. Thus, IRE1α activity is primarily dependent on the magnitude and duration of phosphorylation of its kinase domain. Although the IRE1α-dependent activation of JNKs is considered to initiate a pro-apoptotic response, it also promotes cell survival by triggering cytoprotective autophagy[54,55]. ERK1/2 activation protects against ER stress–induced cell death through induction of prosurvival factors such as GRP78 and MCL-1[19,56,57]. Indeed, in various tumors, constitutive JNK and ER1/2 activation is linked to cell proliferation, survival, and tumor progression[58–60]. In L02 normal liver cells, where the expression level of PRKCSH is low, the interaction between PRKCSH and IRE1α is transient, which leads to a temporary phosphorylation of IRE1α and MAPKs, whereas overexpression of PRKCSH in the same cells results in sustained phosphorylation of IRE1α and MAPKs. In hepatoma cells, where the expression level of PRKCSH is high, interaction of both proteins is relatively lasting and results in sustained phosphorylation of IRE1α and MAPKs. PRKCSH silencing in both cell lines diminished these effects. Although in present study we have not determined how other components of the UPRosome contribute to the association or

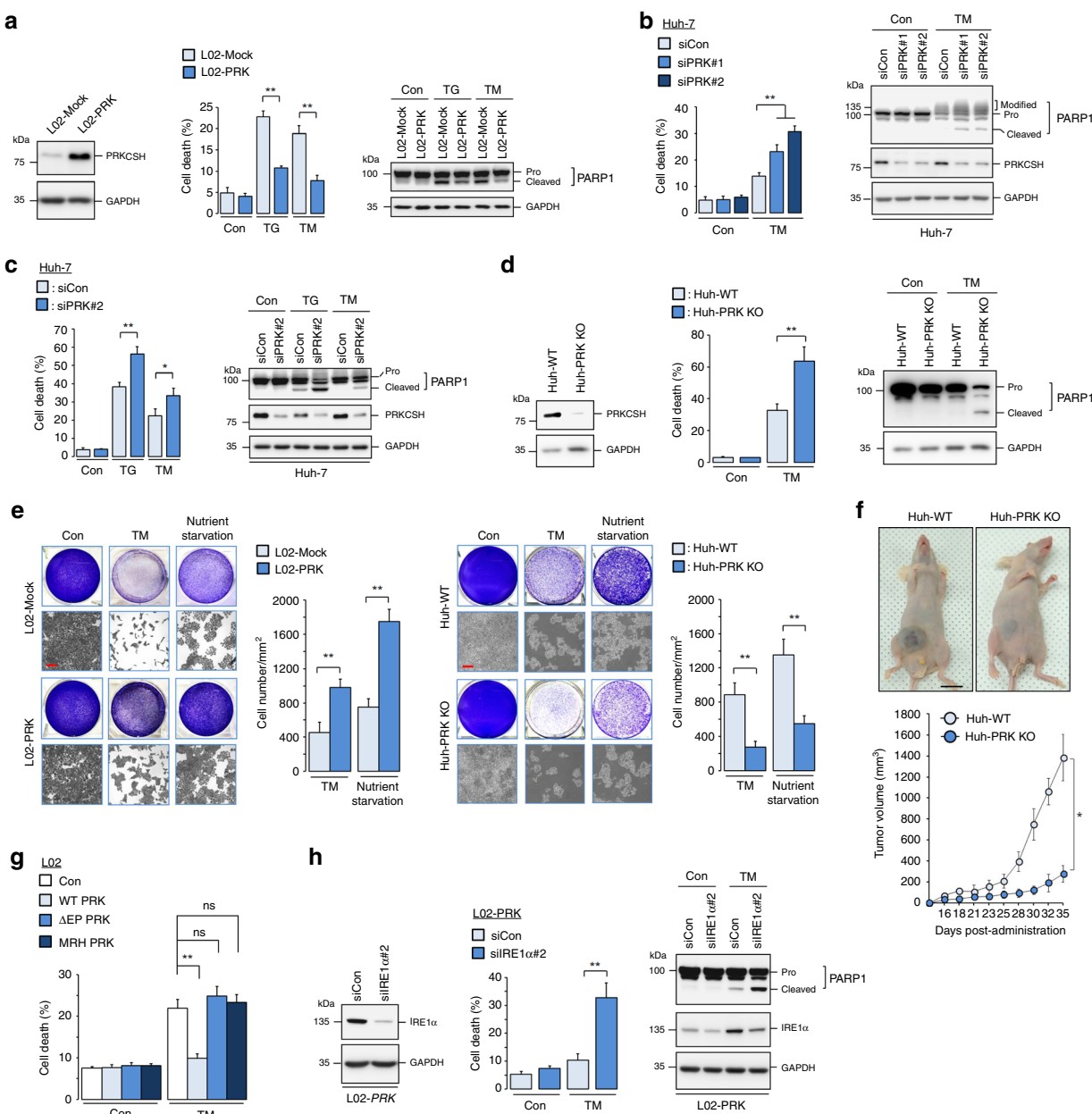

**Fig. 7** PRKCSH mitigates ER stress-induced cell death. **a** Cell death analysis (left) in L02-Mock and L02-PRK cells treated with 10 μg/mL TM or 1 μM TG for 72 h. Cell death was determined from PI staining followed by FACS analysis. Immunoblot analysis of PARP1 cleavage (right) in these cells. **b** Cell death analysis (left) in PRKCSH-silenced Huh-7 cells treated with 10 μg/mL TM for 48 h. Immunoblot analysis of PARP1 cleavage (right) in these cells. **c** Cell death analysis (left) in PRKCSH-silenced Huh-7 cells treated with 10 μg/mL TM or 1 μM TG for 72 h. Immunoblot analysis of PARP1 cleavage (right) in these cells. **d** Cell death analysis (left) in Huh-WT and Huh-PRK KO cells treated with 10 μg/mL TM for 72 h. Immunoblot analysis of PARP1 cleavage (right) in these cells. **e** Cell survival analysis in L02-Mock and L02-PRK cells (left), and Huh-WT and Huh-PRK KO cells (right). Cells were exposed to 10 μg/mL TM or EBSS (for nutrient starvation) for 48 h, and then replated in normal cell culture medium. After 5 days, cell viability was determined by staining with crystal violet. Number of cells per area (mm²) was calculated. Scale bars represent 100 μm. **f** In vivo tumor growth analysis in xenograft nude mice injected with Huh-PRK KO or WT cells. Balb/c nude mice were subcutaneous transplantated with each tumor cells in the right groin (four mice in each group). The volume of tumors was monitored for 35 days. Representative images of tumor-bearing mice (top) and tumor volumes (bottom) were presented. Scale bars represent 10 mm. **g** Cell death analysis in L02 cells transfected with the control, WT PRKCSH, ΔE/P, or MRH mutant plasmids; cells were treated with 10 μg/mL TM for 72 h. **h** Cell death analysis (left) in IRE1α-silenced L02-PRK cells treated with 10 μg/mL TM for 72 h. Immunoblot analysis of PARP1 cleavage (right) in these cells. In cell death analysis and cell survival analysis, data are shown as mean ± SEM from three or four independent experiments. One-way ANOVA; *P < 0.05, **P < 0.01

disassociation of the PRKCSH–IRE1α complex, our results suggest that the endogenous level of PRKCSH in each cell type determines the extent of its interaction with IRE1α and the activity of IRE1α, which is important for adaptation of tumor cells to ER stress.

Activated RNase domain of IRE1α not only induces the splicing of *XBP1* mRNA, but also contributes to the regulated IRE1-dependent decay (RIDD) pathway that cleaves ER-associated RNAs. The splicing of *XBP1* mRNA promotes cell survival, whereas RIDD leads to cell death[61]. IRE1α oligomerization is

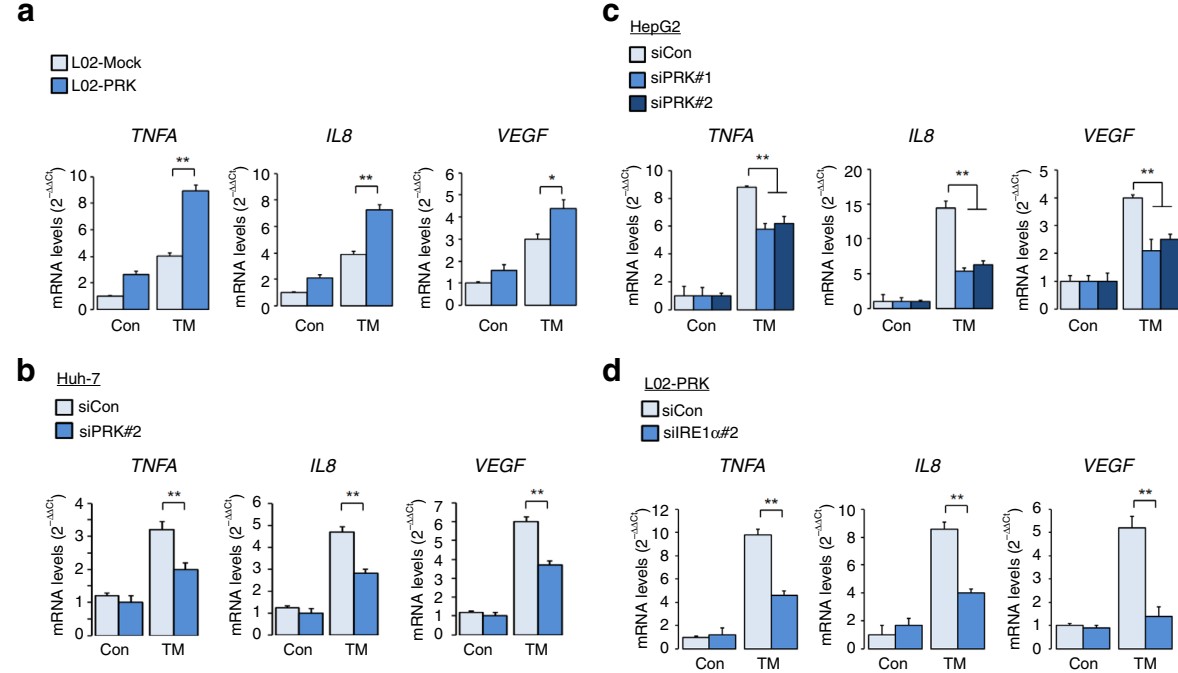

**Fig. 8** PRKCSH enhances expression of tumor-promoting cytokines under ER stress. **a** Quantitative real-time PCR analysis of tumor-promoting cytokines in L02-Mock and L02-PRK cells treated with 10 μg/mL TM for 5 h. **b** Quantitative real-time PCR analysis of tumor-promoting cytokines in PRKCSH-silenced Huh-7 cells treated with 10 μg/mL TM for 5 h. **c** Quantitative real-time PCR analysis of tumor-promoting cytokines in PRKCSH-silenced HepG2 cells treated with 10 μg/mL TM for 5 h. **d** Quantitative real-time PCR analysis of tumor-promoting cytokines in IRE1α-silenced L02-PRK cells treated with 10 μg/mL TM for 5 h. Data are shown as mean ± SEM from three or four independent experiments. One-way ANOVA; *$P < 0.05$, **$P < 0.01$

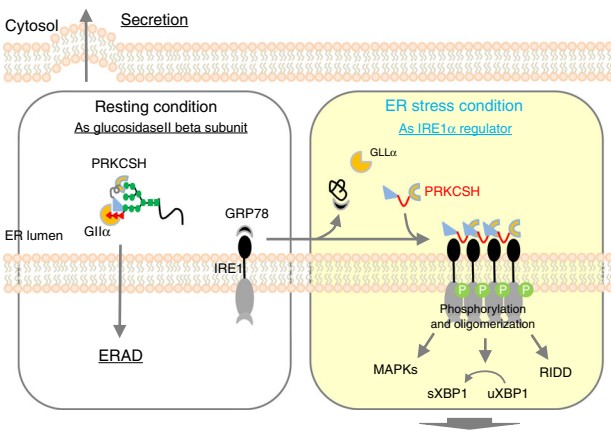

**Fig. 9** A proposed model for the dual function of PRKCSH in ER protein quality control. Under resting conditions, PRKCSH functions as the noncatalytic beta subunit of glucosidase II, which catalyzes glucose trimming from newly synthesized glycoproteins and ensures secretion of properly folded glycoproteins and targeting improperly folded glycoproteins for degradation by the ERAD pathway. This is mediated by the interaction of PRKCSH with glucosidase II alpha subunit (GIIα) through the G2B domain and with mannose residues of glycoproteins through the MRH domain of PRKCSH. Under ER stress, PRKCSH dissociates from the GIIα subunit and subsequently associates with IRE1α through its internal E/P domain, which leads to autophosphorylation and oligomerization of IRE1α followed by RNase-mediated splicing of XBP1 and kinase domain-mediated phosphorylation of MAPKs, thereby selectively promoting activation of the IRE1α signaling branch. Findings from this study are shown on a yellow background, whereas the known function of PRKCSH is shown on a white background

required for maximal splicing of *XBP1* mRNA, which leads to cell survival under ER stress[45,61]. Although it has been reported that nonmuscle myosin heavy chain IIB-mediated oligomerization of IRE1α is caused by direct interaction between both proteins, irrespective of IRE1α phosphorylation[62], it is well-known that autophosphorylation of IRE1α is required for its optimal oligomerization[45]. In the present study, PRKCSH induced phosphorylation of IRE1α followed by its oligomerization. Under ER stress in L02 cells, PRKCSH–IRE1α interaction was transient and the two proteins rapidly dissociated. Nevertheless, IRE1α clustering was promoted after its dissociation from PRKCSH. This fact indicates that association between both proteins does not directly contribute to IRE1α clustering, but increases IRE1α phosphorylation, which in turn promotes IRE1α oligomerization. Consequently, we firmly believe that PRKCSH is required for optimal oligomerization of IRE1α under ER stress, which may paradoxically affect cell fate. In numerous tumors, IRE1α–XBP1 signaling has cytoprotective activity, allowing tumor cells to adapt to ER stress[10–14,63]. PRKCSH expression was positively correlated with the expression of XBP1 target genes such as ER chaperones and ERAD components in various cancer tissues. We found that PRKCSH overexpression in normal liver cells increases the levels of XBP1 splicing and expression of ER chaperones, leading to resistance to ER stress-induced cell death; whereas PRKCSH silencing sensitizes hepatoma cells to ER stress-induced cell death with downregulation of XBP1 splicing and expression of its downstream target genes. Accordingly, IRE1α silencing in L02-PRK cells restores PRKCSH-mediated resistance to ER stress. Overall, the upregulation of PRKCSH in tumor tissues suggests a molecular rationale for altered IRE1α–XBP1 signaling in and adaption of tumor cells to ER stress.

It is accepted that UPR signaling is important for generating malignancy through induction of tumor-promoting factors such as TNF-α, IL-8, and VEGF[5,6]. These factors are also involved in tumor growth because they stimulate proliferation and survival of

tumor cells under stress[5,6]. Indeed, these factors are closely associated with epithelial–mesenchymal transition, angiogenesis, and tumor metastasis[6]. PRKCSH expression was positively associated with the rate of extrahepatic metastasis and TNM stage in HCC tissues. Our findings suggest that PRKCSH contributes to the regulation of ER stress-induced expression of these factors by promoting IRE1α activity. Thus, the PRKCSH–IRE1α signaling axis may be crucial for tumorigenesis through promoting the expression of these factors. We have previously reported that TNF-α induces the expression of PRKCSH, which is involved in a cytokine-mediated response against hepatitis B virus infection[64]. Thus, it is likely that IRE1α activity may be positively linked to PRKCSH expression through the induction of TNF-α expression.

Together with the results of other studies, our data suggest that PRKCSH has a dual function in the ER protein quality control as regulator of GII activity under resting conditions and as a selective regulator of IRE1α under ER stress. Furthermore, we provide a potential mechanism by which different UPR branches can be selectively regulated in tumor cells and how tumor cells can adapt under environmental stress conditions.

## Methods

**Reagents and antibodies**. Propidium iodide (PI)/RNase staining solution (4087) was purchased from Cell Signaling Technology. Tunicamycin (TM; BML-CC104) was obtained from Enzo Life Science. Thapsigargin (TG; 586005) was obtained from Merck. Doxycycline hyclate (D9891) was purchased from Sigma-Aldrich. All stock solutions were prepared according to the manufacturers' instructions.

For immunoblotting, following antibodies were used: PRKCSH (Proteintech, Cat:12148-1-AP, 1:2000), GAPDH (Santa Cruz Biotechnology, Cat:sc-47724, 1:5000), PARP1 (Santa Cruz Biotechnology, Cat:sc-7150, 1:1000), GST (Santa Cruz Biotechnology, Cat:sc-138, 1:1000), XBP1 (Santa Cruz Biotechnology, Cat:sc-7160, 1:1000), PERK (Cell Signaling, Cat:3192, 1:1000), IRE1α (Cell Signaling, Cat:3294, 1:1000), ERK1/2 (Cell Signaling, Cat:4695, 1:1000), phospho ERK1/2 (Cell Signaling, Cat:9101, 1:1000), JNK1/2 (Cell Signaling, Cat:9252, 1:1000), phospho JNK1/2 (Cell Signaling, Cat:9251, 1:1000), GRP78 (Cell Signaling, Cat:3177, 1:1000), ATF4 (Cell Signaling, Cat:11815, 1:1000), Flag M2 (Sigma-Aldrich, Cat: F1804, 1:2000), ATF6 (BioAcademia, Cat:73-500, 1:1000), phospho IRE1α (ser724) (Novus Biologicals, Cat:NB100-2323, 1:1000), anti-mouse conjugated with horseradish peroxidase (Sigma-Aldrich, Cat:SAB3701153, 1:10000), and anti-rabbit antibodies conjugated with horseradish peroxidase (Sigma-Aldrich, Cat:A9169, 1:10000). For immunocytochemistry and immunohistochemistry, we used following antibodies: PRKCSH (Santa Cruz Biotechnology, Cat:sc-46685, 1:200), Calnexin (Cell Signaling, Cat:2679, 1:50), 20S Proteasome α4 (Santa Cruz Biotechnology, Cat:sc-271297, 1:200), Alexa Fluor 546 anti-mouse IgG (ThermoFisher Scientific, Cat:A11030, 1:500), and Alexa Fluor 488 anti-rabbit IgG (ThermoFisher Scientific, Cat:A27034, 1:500). and normal rabbit IgG (2729) were obtained from Cell Signaling Technology.

**Plasmid constructs**. Flag- and Venus-tagged human IRE1α construct (IRE1α-FV) was kindly provided by Dr. Nozomu Kono (University of Tokyo)[46]. Glucosidase II double nickase plasmid (sc-404394-NIC) and control double nickase plasmid (sc-437281) for generation of PRKCSH knockout cells were obtained from Santa Cruz Biotechnology. hIRE1a.pcD was a gift from Dr. Randal Kaufman (Addgene plasmid #21892) and used for IRE1α overexpression. The human PRKCSH gene (Genbank, NM_001001329) was amplified from the hepatoma Huh-7 cell cDNA library. Nontagged PRKCSH cDNA was inserted into the BamH I/Xba I sites of the pcDNA 3.1 vector. Full-length wild-type PRKCSH (WT) and deletion mutants (ΔG2B, ΔE/P, MRH, G2B, E/P, and ΔS/G2B) with a C-terminal Flag-tag were inserted into the BamH I/Xba I sites of the pcDNA 3.1 vector as shown in Fig. 5e, f.

**Cell culture and stable transfection**. HepG2 (HB-8065) and Chang liver (CCL-13) cells were obtained from the American Type Culture Collection. Huh-7 cells (60104) were purchased from the Korean Cell Line Bank. L02 cells (an immortalized normal liver cell line) were kindly provided by Dr. K. H. Lee (Korea Institute of Radiological and Medical Sciences). All cells were cultured in Dulbecco's modified Eagle's medium (DMEM) (Welgene, LM001-05) supplemented with 10% fetal bovine serum and 100 U/mL of penicillin and streptomycin at 37 °C in a 5% CO2 incubator.

Cells stably overexpressing PRKCSH (L02-PRK cells) were generated by transfecting L02 cells with a non-tagged PRKCSH construct. L02-PRK cells were grown in DMEM supplemented with G418 (200 μg/mL) (InvivoGen, ant-gn-1). Glucosidase II double nickase plasmids or control double nickase plasmid were transfected into L02 cells. Stable knockout cells (L02-PRK KO and Huh-PRK KO) were selected in DMEM supplemented with puromycin (500 or 250 ng/mL) (Santa Cruz Biotechnology, sc-205821), and knockout was confirmed by PRKCSH

immunoblotting. All maintained cells were verified to be free of mycoplasma contamination by polymerase chain reaction (PCR).

**Gene silencing by siRNA**. All small interfering RNA (siRNA) duplexes were synthesized by ST Pharm Co. (Seoul, Republic of Korea). Two siRNAs targeting PRKCSH (siPRK #1, 5′-GGA AGA AGU CUC UGG AAG ATT-3′ and siPRK #2, 5′-GGA AGA AGA GGC UGA AGA ATT-3′) were used. One siRNA targeting IRE1α (siIRE1α #2, 5′-GAA UCC UCU ACA UGG GUA AAA AGC ATT-3′) was used. One nonspecific scrambled control siRNA (siCon, 5′-AUG AAC GUG AAU UGU UCA ATT-3′) was used. Cells were transfected with siRNAs (final concentration, 40 nM) using Lipofectamine 2000 (ThermoFisher Scientific, 11668-019) according to the manufacturer's instructions. At 48 h post-transfection, cells were subjected to immunoblotting to confirm target expression.

**Cell treatment with ER stressors**. Individual PRKCSH constructs or siRNAs were transiently transfected by using Lipofectamine 2000. At 48 h post-transfection, cells were treated with (final concentrations) 1 μM TG or 10 μg/mL TM for the indicated times. Stable cell lines (L02-PRK, L02-PRK KO, or Huh-PRK KO) were cultured in DMEM without G418 or puromycin for 7 days before treatment with an ER stressor, and were then seeded into a 6-well plate (2 × 10^5 cells/mL). After 24 h, medium was changed to fresh medium with TG or TM for the indicated times. Glucose deprivation was induced by replacing medium with fresh DMEM without glucose (Welgene, LM001-56) supplemented with 10% dialyzed fetal bovine serum (Welgene, S001-08) for the indicated times.

**Gene expression and survival data from cancer tissues**. Datasets of gene expression in human tumors were obtained from TCGA. Gene Expression Profiling Interactive Analysis (GEPIA), a web-based tool (http://gepia.cancer-pku.cn/), was used for analysis of differential expression of *PRKCSH* mRNA, creation of the human anatomy clipart, analysis of correlation with the expression of XBP1 downstream target genes, and patient survival analysis[65].

Other gene expression datasets for liver cancers (GSE25097 and GSE20140), colon cancer (GSE10950), gastric cancer (GSE13861), breast cancer (GSE24124), and lung cancer (GSE27262) were downloaded from the NCBI GEO database. A profile graph was used to extract the expression values of specific genes by entering the corresponding identifier from the ID column. For survival analysis of patients with liver cancer, a gene expression dataset for hepatocellular carcinoma samples (E-TABM-36) was downloaded from the ArrayExpress database of EMBL-EBI. Processed data were used to extract the expression values of the *PRKCSH* gene and clinical data.

**IHC analysis of liver tissues**. Liver tissue array slides were purchased from SuperBioChips Laboratories and used to determine the levels of PRKCSH by IHC analysis. The slides contained non-tumor (CSN3, 59 cases) and tumor (CS3, 58 cases) liver tissue specimens. Clinicopathological information is available at the manufacturer's website (http://www.tissue-array.co.kr/). The slides were baked at 60 °C for 30 min, deparaffinized with xylene and dehydrated with ethanol. The slides were subjected to antigen retrieval and incubated with blocking solution to prevent nonspecific antibody binding, followed by incubation with anti-PRKCSH antibody overnight at 4 °C. After counterstaining with hematoxylin, the sections were dehydrated and mounted. Staining intensity of the PRKCSH protein was measured using the NIH ImageJ software with the IHC Profiler plugin (http://rsb.info.nih.gov/ij/)[66]. Staining intensity of PRKCSH is shown as the inverted median pixel value (IMPV). The PRKCSH-positivity was presented as strong positive (>121 IMPV), positive (71–121 IMPV), weak positive (30–70 IMPV), and negative (<30 IMPV) (Supplementary Fig. 1c).

**Immunoblot analysis**. Cells were lysed with sodium dodecyl sulphate (SDS) lysis buffer (100 mM Tris-HCl, pH 6.8, 10% glycerol, and 1% SDS) supplemented with protease inhibitor cocktail (ThermoFisher Scientific, 78441). Protein concentration was determined using the BCA protein assay kit (ThermoFisher Scientific, 23225). Cell lysates were boiled in 1× sample buffer (10 mM Tris–HCl pH 6.8, 1% SDS, 5% glycerol, 0.05% bromophenol blue, and 1% β-mercaptoethanol) for 5 min. Proteins were separated on SDS-PAGE and electrotransferred to Immobilon-P membranes (Merck, IPVH00010). Immunoblotting with individual primary antibodies was performed as recommended by the suppliers. Signals were detected using an LAS-4000 Luminescent Image Analyzer (GE Healthcare Bio-Sciences) and their intensity was determined by densitometry using the Multi Gauge software (Fujifilm). The intensity of each protein signal was normalized to that of total protein or GAPDH. The control value was set to 1.0 and protein intensity was represented as a ratio to control. Uncropped images of all immunoblots are provided in Supplementary Fig. 10.

**RNA extraction and quantitative PCR**. Total RNA was isolated using the Trizol reagent (Sigma-Aldrich, T9424) and was reverse transcribed using M-MLV reverse transcriptase (IntRon Biotechnology, 27032) and an oligo-dT primer according to the manufacturer's instructions. Quantitative PCR was performed on a 7500 real-time PCR System (Applied Biosystems) with SYBR Green PCR Master Mix

(ThermoFisher Scientific, 4309155) following the manufacturers' protocols. To investigate splicing pattern of XBP1 mRNA, we also performed semiquantitative PCR on an XP Thermal Cycler System (BIOER Technology, Hangzhou, China). Uncropped images of all agarose gels are provided in Supplementary Fig. 10. All primer sequences are listed in Supplementary Table 2.

**Imaging of IRE1α foci**. To monitor the IRE1α oligomerization level, IRE1α foci in cells transfected with IRE1α-FV were examined by immunofluorescence microscopy[46]. For PRKCSH silencing, cells were transfected with siPRK or siCon. At 24 h post-transfection, cells grown on coverslips were transfected with IRE1α-FV. At 24 after the second transfection, cells were treated with doxycycline (5 μg/mL) for 24 h to induce expression of IRE1α-FV, followed by treatment with TM for 4 h. Cells were fixed with 3.7% paraformaldehyde (Electron Microscopy Sciences, 15710) for 15 min at room temperature. Knockdown of PRKCSH in this cell was evaluated by immunocytochemistry with anti-PRKCSH antibody. For stable L02-PRK cells, cells grown on coverslips were transfected with IRE1α-FV. At 16 h post-transfection, cells were treated with doxycycline (5 μg/mL) for 24 h to induce expression of IRE1α-FV, followed by treatment with TM for 4 h. Cells were fixed with 3.7% paraformaldehyde for 15 min at room temperature. Oligomerization of IRE1α was observed using a Carl Zeiss Axiovert 200 fluorescence microscope and analyzed with the software supplied by the manufacturer (Carl Zeiss). The number of IRE1α foci per cell was determined by counting more than 100 cells with fluorescent IRE1α. Data were presented as percentage of cells with IRE1α foci among all cells with fluorescent IRE1α.

To examine the association of free or oligomerized IRE1α with the proteasome, cells grown on coverslips were transfected with IRE1α-FV. At 16 h post-transfection, cells were treated with doxycycline (5 μg/mL) for 24 h to induce IRE1α-FV expression, followed by treatment with TM for 4 h. Cells were fixed with 3.7% paraformaldehyde for 15 min at room temperature. After washing with PBS, cells were probed with anti-20S proteasome α4 antibody overnight at 4 °C. After three washes with PBS, the cells were incubated with Alexa Fluor 546-conjugated anti-mouse IgG secondary antibody at room temperature for 40 min.

**Immunocytochemistry**. To determining PRKCSH expression upon its knockdown, cells were seeded on coverslips and transfected with siPRK or siCon. At 48 h post-transfection, cells were fixed with 3.7% paraformaldehyde for 15 min at room temperature. After washing with PBS, cells were probed with anti-PRKCSH antibody at 4 °C overnight. After three washes with PBS, the cells were incubated with Alexa Fluor 546-conjugated anti-mouse IgG secondary antibody at room temperature for 40 min. Fluorescence images were obtained using a Carl Zeiss Axiovert 200 microscope and analyzed by using the supplier's software. The intracellular localization of WT PRKCSH and its mutants were determined by indirect immunofluorescence analysis using anti-Calnexin (Cell signaling Technology, 2679) anti-Flag M2 antibodies (Sigma-Aldrich), or anti-PRKCSH (Santa Cruz Biotechnology).

**Subcellular fractionation**. Cells were lysed in buffer A (25 mM HEPES, pH 7.4, 50 mM KCl, 1 mM EDTA, 0.5% NP-40, 1 mM DTT) supplemented with a protease inhibitor cocktail (ThermoFisher Scientific). After incubation on ice for 30 min, the samples were separated by centrifugation. Supernatants were collected as cytosolic fractions. The pellets were washed with buffer B (buffer A without NP-40) and the recovered nuclear fraction was washed with buffer C (buffer B containing 450 mM KCl and 50% glycerol). Tubulin (Cell Signaling Technology, 2144) was used as a cytosolic marker and lamin A/C (Cell Signaling Technology, 2032) as a nuclear marker.

**Co-IP assay**. Nontransfected cells or cells transiently transfected with individual constructs were lysed with immunoprecipitation lysis buffer (50 mM Tris-HCl, pH 7.2, 10 mM NaCl, 1% NP-40 [Sigma-Aldrich, NP40S], and a protease inhibitor cocktail [ThermoFisher Scientific, 78441]). The lysate was precleared with protein A agarose (Sigma-Aldrich, 11134515001) at 4 °C for 1 h and incubated with anti-IRE1α (Cell Signaling, 1:50), anti-Flag M2 (Sigma-Aldrich, 1:50), or normal rabbit IgG antibody (Cell Signaling, Cat:2729, 1:50) at 4 °C overnight, and then with protein A agarose at 4 °C for 4 h. The agarose was washed three times with lysis buffer and boiled in 1× sample buffer. Boiled samples were subjected to immunoblot analysis.

**GST affinity isolation assay**. N-terminally GST-tagged PRKCSH (LifeSpan BioSciences, LS-G22189) or C-terminally GST-tagged PRKCSH (Abnova, H00005589-P02) was immobilized on glutathione-Sepharose 4B beads (GE Healthcare Bio-Sciences, 17-0756-01)[67]. Recombinant human IRE1α (OriGene Technologies, TP315023) was incubated with immobilized GST-PRKCSH proteins in TEN buffer (20 mM Tris-HCl, pH 7.5, 200 mM NaCl, 1 mM EDTA, and protease inhibitor cocktails) at 4 °C overnight. The beads were washed four times with TEN buffer, eluted with SDS-PAGE sample buffer, and analyzed by immunoblotting with antibodies against each protein.

**ER stress-induced cell death assay**. Apoptotic cell population was determined by evaluating the DNA content of hypodiploid cells by the PI flow cytometric assay[66,68]. Cells were seeded, and (or) transfected with individual siRNAs for 48 h. Cells were treated with TM or TG for 48 or 72 h. Cells were fixed with cold 70% ethanol and treated with PI/RNase staining solution as described in the

manufacturer's instructions. Cells were analyzed with a FACSCalibur flow cytometer (BD Biosciences), and cellular DNA content and forward scatter were analyzed with FCS Express 6 Plus software (De Novo Software) or WinMDI2.9 software. To determine PARP1 cleavage, cells were prepared as above, and then immunoblot analyses were performed with anti-PARP1 antibody as recommended by the supplier.

**Cell survival and in vivo tumor growth assay**. Cell survival assay was performed for determining the long-term overall survival of L02-PRK or Huh-PRK KO cells[69,70]. Totally, $2 \times 10^5$ cells/mL were seeded in 6-well plate and treated with 10 μg/ml TM, or exposed to Earle's balanced salt solution for 48 h. Cells were washed and trypsinized. A total of $1 \times 10^4$ cells/mL were replated into 6-well plate and cultured for 5 days in complete DMEM media supplemented with 10% fetal bovine serum. Cells were washed three times in PBS, fixed with 3.7% paraformaldehyde, stained with crystal violet for 30 min, and then washed three times with water. Survival rate was calculated by cell number/mm$^2$.

In vivo tumor growth experiment was performed in accordance with the guidelines of the Committee for Animal Experiments of Konkuk University. All animal protocols were approved by the Institutional Animal Care and Use Committee of Konkuk University (Protocol no. KU19032). Male BALB/c nude mice (6–7 weeks old, 25 g) were purchased from Orient Bio (Seongnam, Republic of Korea). The Huh-PRK KO or WT cells ($5 \times 10^6$ cells per 0.1 mL Hank's Balanced Salt Solution) were injected subcutaneously in the right groin. The mice were monitored daily and tumor sizes were measured every 2–3 days by a digital caliper, and tumor volumes were calculated using the formula $V = \pi/6 \, (L \times W^2)$, where $W$ and $L$ are tumor width and length, respectively.

**Statistical analysis**. All experiments were repeated at least three times. Data were expressed as mean ± SEM. Differences between two groups were analyzed by the Student's $t$ test. Multigroup comparisons were performed by one-way analysis of variance. IHC staining of tissue array was assessed using the chi-square test. Prognoses for patients with hepatocellular carcinoma were estimated by using Kaplan–Meier survival analyses with the log-rank test. Correlations between PRKCSH and UPR gene expression in various cancer tissues were assessed using Pearson's rank correlation coefficient. The differences between the groups in cancer tissues data were compared using the unpaired or paired two-tailed Student's $t$ test. Statistical analyses were performed and graphs were plotted using the GraphPad Prism software (version 6, GraphPad Software, Inc). $P$ values <0.05 were considered statistically significant.

## Data availability

All data supporting the findings of this study are available within the article and its Supplementary Information files and from the corresponding author upon reasonable request. TCGA datasets of gene expression in human tumors are available from GEPIA web server. Additional gene expression datasets were obtained for liver cancers (GSE25097 and GSE20140), colon cancer (GSE10950), gastric cancer (GSE13861), breast cancer (GSE24124), and lung cancer (GSE27262). A reporting summary for this article is available as a Supplementary Information file.

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

## Acknowledgements

The authors gratefully acknowledge Dr. Nozomu Kono (University of Tokyo) for providing the IRE1-FV construct. We also gratefully acknowledge Dr. Randal Kaufman for providing the hIRE1a.pcD construct. This study was supported by the National Research Foundation of Korea (NRF) grants funded by the Korea government (NRF-2015R1D1A1A01057281, NRF-2017R1A2B3006335, NRF-20181D1A1B07044129, and NRF-2016R1A5A2012284), and by the Korea Health Technology R&D Project through the Korea Health Industry Development Institute (KHIDI), funded by the Ministry of Health & Welfare (No. HI17C0874).

## Author contributions

G.C.S. and K.H.K. designed the experiments and analyzed the data; G.C.S., S.U.M., H.S. C., and H.D.H. performed the experiments; H.S.K. analyzed the bioinformatic and statistical data. All authors discussed the results and analysis; G.C.S. and K.H.K. co-wrote the paper, with contributions from all authors.

## Additional information

**Competing interests:** The authors declare no competing interests.

