## [Peer Review File · Nature Communications]

Reviewers' comments:

Reviewer #1 (Remarks to the Author):

The manuscript by Shin et al., describes studies suggesting a novel role to the PRKCSH/Hepatocystin protein, a non-catalytic subunit of glucosyltransferase II as a modulator of the function of the major UPR effector IRE1. A series of biochemical analyses demonstrates that PRK can bind and modulate the kinase activity for IRE1alpha. Coupled with database mining analysis demonstrating increased levels of PRK in tumors compared to normal tissues and data showing that increased PRK levels promote a more pro-survival activity of IRE1alpha against cytotoxic ER stress, the authors propose that PRK may have a pro-tumorigenic role in protect tumor cells from ER stress-induced death. The work presented here contains some novel aspects, particularly the finding and characterization of interactions between IRE1alpha and PRK, which include elegant analysis of the domains involved in this interaction. However, there are significant gaps in the overall rationale linking PRK upregulation with a more aggressive tumor cell phenotype, which are exacerbated by the lack of any in vivo tumor data testing the central hypothesis. These issues are compounded by the modest effects of upregulated or downregulated PRK has on IRE1a activity.

Major concerns:

1. The analysis of PRK expression in human tumors shows statistically significant (based on P value) data for mRNA. However, the magnitude of the differences in expression is marginal and unclear if it is biologically meaningful. Moreover, the use of select datasets of mRNA analyses in the specific tumor types (as opposed to complete data sets from TCGA), raises concerns of selection bias. Similar concerns are raised regarding the levels of PRK protein, shown in Fig. 1C, which appear to be very modest in magnitude. Similar issues are observed in Fig. 3g, where the correlation of mRNA levels between PRK and targets of the Ire1/XBP1 pathway are poor (e.g., r values < 0.3).
2. For the most experiments dealing with overexpression of PRK in untransformed cells (e.g., Figs. 2b, 2b, 2f, 3a, 4a etc), or in some cases of PRK knockout (Fig. 2e), the magnitude of the observed effects is marginal at best (i.e., from 1 to 1.5-fold, or from 2-fold to 3-fold increase in splicing of XBP1). What is of particular concern here, is that these changes arise in response to extreme stress conditions, either with tunicamycin treatment or complete absence of glucose, which are unlikely to be physiologically relevant.
3. The knockdown of PRK with shRNA is not complete and perhaps this is a reason some of the results are underwhelming. CRISPR-mediated knockouts should be attempted.
4. The co-immunoprecipitation experiments in Fig. 5 lack important controls. First, an IgG control Ab needs to be included to determine non-specific IP activity (panels 5a-5e). Second, the reverse IP with anti-PRK antibody should demonstrate activity in bringing down IRE1a. Moreover, the quality of Fig. 5f is poor and the blot should be repeated.
5. In Fig. 6a, does expression of the PRK mutant $\Delta E/P$ or MRH constructs affect survival under ER stress conditions?
6. The cytoprotective data demonstrated by immunoblots against cleaved PARP are also very weak. Clonogenicity assays need to be performed to determine if long-term overall survival (including necroptosis, ferroptosis, as well as senescence, necrosis etc) is affected.
7. Since all of the ER stresses used almost exclusively rely on large doses of pharmacological agents such as tunicamycin or thapsigargin, it is critical that in vivo tumor growth experiments, where transformed hepatocytes are exposed to physiological stress (e.g., hypoxia, low nutrient availability).

Reviewer #2 (Remarks to the Author):

The UPR responds to intrinsic and extrinsic stresses that disrupt protein folding in the endoplasmic

reticulum. It is now apparent that tumor survival pathways utilize the UPR as a pro-survival strategy and as such there is interest in drugging PERK and IRE1 pathways. While many stresses activate all 3 branches of the UPR, ATF6, PERK and IRE1, conditions have been reported wherein one pathway is selectively activate, although this is not observed in tumorigenesis. Indeed, protein misfolding reduces binding of BiP to the highly homologous luminal domains of all three UPR transducers permitting oligomerization of PERK and IRE (ER to golgi transport of ATF6), thereby eliciting activation. Whether pathways exist and how they regulate activation of a single branch of the UPR is poorly defined and molecular elucidation of such pathways would be of significant interest. The work by Shin et al define PRKCSH (a component of ER localized glucosidase II as a direct and specific activator of IRE. While the data does support activation of this pathway, there are gaps. For example, data for IRE specificity over other branches is not strong. It is unclear why PRKCSH would bind selectively to IRE when the luminal domains are highly homologous and in fact interchangeable. In addition, interpretation of results and quality of some key experiments needs to be re-evaluated. Specific points for consideration are listed below.

1. What happens to glucosidase activity during ER stress? I assume that PRKCSH is being redirected and thus would limit activity. Alternatively, while the authors show direct activation, of this subunit, is the entire enzyme complex recruited and is glucosidase activity needed for activation of IRE?
2. Looking at the data, it is clear that PRKCSH is not required for activation; rather it seems to provide a boost in signaling. Indeed, it appears that the kinetics of Xbp1 splicing are unaltered and differences in spliced Xbp are not apparent until 5 hours suggesting that PRKCSH modifies the response but is not necessary.
3. With respect to the comment above, the dose of tunicamycin used throughout are very high. A dose response and use of a lower dose of TM would be more compelling throughout.
4. In the beginning of the results, the authors never define what PRK cells are either in text or figure legends.
5. Figure 2 and 3 provide evidence for PRKCSH in increasing Xbp1 splicing and later that it increases IRE1 auto-p. However, the two are never tied together. The authors should determine whether Ire1 is necessary for PRKCSH-dependent effects on Xbp1 (knockdown of Ire or Ire1 inhibitor). It would be of equal interest to determine whether overexpression of Ire1 (which increases its activity through forced oligomerization) would overcome the effect of PRKCSH knockout or knockdown. The issues with the dose of TM, the kinetics of the response, this data would help support that the impact is directly on Ire1.
6. The specificity for Ire1 is not convincing given high homology of all UPR transducers in their luminal domains. In the supplemental data, it is not sufficient to only show PERK-p. The authors need to perform the same time course as with Ire⁻¹ and XBP1, and examine downstream signaling; p-eIF2, ATF4 and CHOP. These experiments should also be performed in PRKCSH k/o cells for clarity.
7. The title of the section "PRKCSH directly activates Ire1" is misleading and not accurate. The data demonstrate it binds, but that binding correlates with increased auto-P. However, no data demonstrate activation and PRKCSH is clearly not required for binding since there is activity in its absence.
8. The experiments interpreted to demonstrate that PRKCSH increases oligomerization could reflect formation of inclusions at the proteasome that reflects overexpression. These experiments do not in fact establish oligomerization. Use of differential tagging would be one possible approach to compliment and control for this issue.
9. Defining the E/P domain as necessary for binding is excellent. However, it is disappointing that the authors did not: 1) demonstrate specificity for Ire; 2) determine whether it is sufficient to activate Ire1 and do so selectively.
10. With regard to specificity, if PRKCSH indeed targets Ire1 specifically, interchanging luminal domains should alter or switch specificity and provide direct support for the authors hypothesis.

Reviewer #1 (Remarks to the Author):

The manuscript by Shin et al., describes studies suggesting a novel role to the PRKCSH/Hepatocystin protein, a non-catalytic subunit of glucosyltransferase II as a modulator of the function of the major UPR effector IRE1. A series of biochemical analyses demonstrates that PRK can bind and modulate the kinase activity for IRE1 α . Coupled with database mining analysis demonstrating increased levels of PRK in tumors compared to normal tissues and data showing that increased PRK levels promote a more pro-survival activity of IRE1 α against cytotoxic ER stress, the authors propose that PRK may have a pro-tumorigenic role in protect tumor cells from ER stress-induced death. The work presented here contains some novel aspects, particularly the finding and characterization of interactions between IRE1 α and PRK, which include elegant analysis of the domains involved in this interaction. However, there are significant gaps in the overall rationale linking PRK upregulation with a more aggressive tumor cell phenotype, which are exacerbated by the lack of any in vivo tumor data testing the central hypothesis. These issues are compounded by the modest effects of upregulated or downregulated PRK has on IRE1 α activity.

Major concerns:

1. The analysis of PRK expression in human tumors shows statistically significant (based on P value) data for mRNA. However, the magnitude of the differences in expression is marginal and unclear if it is biologically meaningful. Moreover, the use of select datasets of mRNA analyses in the specific tumor types (as opposed to complete data sets from TCGA), raises concerns of selection bias. Similar concerns are raised regarding the levels of PRK protein, shown in Fig. 1C, which appear to be very modest in magnitude. Similar issues are observed in Fig. 3g, where the correlation of mRNA levels between PRK and targets of the Ire1/XBP1 pathway are poor (e.g., r values < 0.3).

Response: Thank you for commenting. Considering this comment, we performed the expression analysis of PRKCSH in various human tumors using complete data sets from TCGA. We analyzed these data sets using GEPIA (Gene Expression Profiling Interactive Analysis, <http://gepia.cancer-pku.cn/>), a web-based tool that delivers fast and customizable functionalities based on TCGA data. By using GEPIA, we performed differential expression analysis, expression profiling plotting, correlation analysis, and patient survival analysis. In the revised manuscript, the data clearly show that PRKCSH expression is increased in various cancer tissues such as GBM (glioblastoma multiforme), ESCA (esophageal carcinoma), DLBC (lymphoid neoplasm diffuse large B-cell lymphoma), THYM (thymoma), LIHC (liver hepatocellular carcinoma), PAAD (pancreatic adenocarcinoma), STAD (stomach adenocarcinoma), and SKCM (skin cutaneous melanoma) (**Fig. 1a, 1b, Supplementary Fig. 1a**; described in the Results and Methods sections of the revised manuscript, **pages 7 and 27**).

In addition, we performed further analysis of PRKCSH protein expression in HCC tissues and analyzed the IHC data by using the NIH-ImageJ software with the IHC Profiler plugin (<http://rsb.info.nih.gov/ij/>) as described previously (**Autophagy 2016, 12: 2451-2466; PLoS One 2014, 9: e96801**). Staining intensity of the PRKCSH protein is now presented as IMPV (inverted median pixel value; **Supplementary Fig. 1c**). In the revised manuscript, the data clearly show that level of the PRKCSH protein is higher in liver cancer tissues than in normal liver tissues (**Fig. 1d**; described in the Results and Methods sections of the revised manuscript, **pages 7 and 28**).

We also further analyzed the patient survival curves using the data sets from TCGA and demonstrate that PRKCSH expression is associated with both the overall survival and disease-free survival of patients with liver hepatocellular carcinoma and skin cutaneous melanoma (**Fig. 1g**; described in the Results of the revised manuscript, **page 8**).

Finally, we analyzed the correlation between mRNA levels of PRKCSH and the IRE1/XBP1

target genes using complete data sets from TCGA by GEPIA web tool. We demonstrate that the expression level of PRKCSH is clearly correlated with those of XBP1 downstream target genes (**Fig. 3h and Supplementary Fig. 4**; described in the Results of the revised manuscript, **page 10**).

2. For the most experiments dealing with overexpression of PRK in untransformed cells (e.g., Figs. 2b, 2b, 2f, 3a, 4a etc), or in some cases of PRK knockout (Fig. 2e), the magnitude of the observed effects is marginal at best (i.e., from 1 to 1.5-fold, or from 2-fold to 3-fold increase in splicing of XBP1). What is of particular concern here, is that these changes arise in response to extreme stress conditions, either with tunicamycin treatment or complete absence of glucose, which are unlikely to be physiologically relevant.

Response: Tunicamycin treatment and glucose deprivation are commonly used as experimental conditions for the induction of ER stress and are generally accepted as reliable *in vitro* models in this research field (**Science 2007, 318:944-949; Proc Natl Acad Sci USA 2017, 114:3192-3197**). Accordingly, numerous research groups have used these stress inducers for functional studies of various genes associated with ER stress response. The previously reported changes in the magnitude of XBP1 splicing levels under these stress conditions, which have been accepted as significant (**Nature 2002, 415:92-96; PLoS Biol 2010, 6:e1000410; Developmental Cell 2012, 23:1141-1152**), are similar to our results. Thus, we think that our data is sufficient for explaining the significant differences in XBP1 splicing and MAPK activation between control and PRKCSH-modulated groups under these ER stress conditions. Our data shows that PRKCSH is boosts IRE1 α signaling, but is not necessary for its triggering.

Unlike TM, glucose starvation (along with hypoxia, oxidative stress, and nutrient starvation) is generally recognized as one of the most representative physiological stress conditions for

in vitro modeling in the field of ER stress research (**Cancer Res.** 2007, 67:6700-6707; **Nature** 2014, 508:103-107; **Nat Rev Mol Cell Biol.** 2002, 3:411-421).

3. The knockdown of PRK with shRNA is not complete and perhaps this is a reason some of the results are underwhelming. CRISPR-mediated knockouts should be attempted.

Response: Following the reviewer's comment, we established the Huh-PRK KO cell line using the CRISPR system and performed experiments using this cell line. In the revised manuscript, we provide additional evidence that PRKCSH is involved in promoting ER stress-induced XBP1 splicing (**Fig. 3e**; described in the Results of the revised manuscript, **page 10**), expression of XBP1 target genes (**Fig. 3g**; described in the Results of the revised manuscript, **page 10**), and IRE1 α phosphorylation (**Fig. 4f**; described in the Results of the revised manuscript, **page 12**), but not in the activation of the PERK pathway (**Supplementary Fig. 5d and Supplementary Fig. 5e**; described in the Results of the revised manuscript, **page 11**). Subsequently, by using this PRK-KO cell line, we confirmed that PRKCSH is associated with resistance to ER stress-induced cell death (**Fig. 6d and 6e**; described in the Results of the revised manuscript, **page 16**). Overall, we obtained clearer results with the knockout cells than with the knockdown cells.

4. The co-immunoprecipitation experiments in Fig. 5 lack important controls. First, an IgG control Ab needs to be included to determine non-specific IP activity (panels 5a-5e). Second, the reverse IP with anti-PRK antibody should demonstrate activity in bringing down IRE1 α . Moreover, the quality of Fig. 5f is poor and the blot should be repeated.

Response: We appreciate these important comments. We performed all the control co-immunoprecipitation experiments using control IgG Ab and included the results in the revised

manuscript (**Fig. 5a, 5b, 5g, and Supplementary Fig. 5f**). We also performed co-immunoprecipitation experiments using anti-PRKCSH antibody and IRE1-Flag-transfected cells (**Fig. 5c**). The experiment shown in Fig. 5f in the original manuscript was replaced with better quality blots (**Fig. 5c**). Overall, the newly added data provides clear evidence of the interaction between PRKCSH and IRE1 α .

5. In Fig. 6a, does expression of the PRK mutant Δ E/P or MRH constructs affect survival under ER stress condition?

Response: Following this comment, we performed cell death assay using WT, Δ E/P, and MRH mutant constructs under ER stress. Our data provide additional evidence that the E/P domain is required for PRKCSH-mediated resistance to ER stress-induced cell death (**Fig. 6f**; described in the Results of the revised manuscript, **page 16**).

6. The cytoprotective data demonstrated by immunoblots against cleaved PARP are also very weak. Clonogenicity assays need to be performed to determine if long-term overall survival (including necroptosis, ferroptosis, as well as senescence, necrosis etc) is affected.

Response: As the reviewer pointed out, ER stress-induced apoptotic cell death in liver cell lines used in the present study was relatively weak compared to that in murine embryonic fibroblasts (MEFs) described in previous reports (**Cell Death and Differentiation 2012, 19:1013-1026; EMBO J. 2011, 30:4465-4478**). Indeed, our liver cell lines did not show a notable uptake of PI (indicator of necrotic cell death) until 72h after treatment with tunicamycin or thapsigargin, unlike MEFs described in previous reports (**Cell Death and Differentiation 2012, 19:1013-1026; EMBO J. 2011, 30:4465-4478**). Therefore, we presumed that these liver cell lines are relatively resistant to ER stress-induced cell death.

Nevertheless, according to the reviewer's comment, we performed clonogenic cell survival assay using L02-PRK overexpression and Huh-PRK KO cells after treatment with TM (**Fig. 6e**) as described in previous reports (**Cell Death and Differentiation 2012, 19:1013-1026; EMBO J. 2011, 30:4465-4478**). In the revised manuscript, we provide clear evidence that PRKCSH contributes to cytoprotection against ER stress-induced cell death (**Fig. 6e**; described in the Results and Methods sections of the revised manuscript, **pages 16 and 32**).

7. Since all of the ER stresses used almost exclusively rely on large doses of pharmacological agents such as tunicamycin or thapsigargin, it is critical that *in vivo* tumor growth experiments, where transformed hepatocytes are exposed to physiological stress (e.g., hypoxia, low nutrient availability).

Response: Treatment with 5–10 µg/ml tunicamycin has been generally used to induce ER stress-mediated cell death and to assess the cytoprotective function of some molecules against ER stress (**J Cell Sci. 2016, 129:2317-2328; Oncotarget 2016, 22:76827-76839; Mol Biol Cell. 2014, 25:1411-1420; Blood 2009, 113:626-634**). Nevertheless, we also had concerns raised in the reviewer's comment, namely whether PRKCSH has a cytoprotective role under physiological stress conditions such as hypoxia or very low nutrient availability. Considering that there is no suitable *in vivo* tumor model for ER stress research, we mimicked the *in vivo* tumor growth by performing the clonogenic cell survival assay under nutrient-starvation conditions using L02-PRK and Huh-PRK KO cells, as described in previous reports (**Cell Death and Differentiation 2012, 19:1013-1026; EMBO J. 2011, 30:4465-4478**). In the revised manuscript, we provide additional evidence that PRKCSH may contribute to tumor cell resistance against physiological stresses (**Fig. 6e**; described in the Results of the revised manuscript, **page 16**).

Reviewer #2 (Remarks to the Author):

The UPR responds to intrinsic and extrinsic stresses that disrupt protein folding in the endoplasmic reticulum. It is now apparent that tumor survival pathways utilize the UPR as a pro-survival strategy and as such there is interest in drugging PERK and IRE1 pathways. While many stresses activate all 3 branches of the UPR, ATF6, PERK and IRE1, conditions have been reported wherein one pathway is selectively activate, although this is not observed in tumorigenesis. Indeed, protein misfolding reduces binding of BiP to the highly homologous luminal domains of all three UPR transducers permitting oligomerization of PERK and IRE (ER to golgi transport of ATF6), thereby eliciting activation. Whether pathways exist and how the regulate activation of a single branch of the UPR is poorly defined and molecular elucidation of such pathways would be of significant interest. The work by Shin et al define PRKCSH (a component of ER localized glucosidase II as a direct and specific activator of IRE. While the data does support activation of this pathway, there are gaps. For example, data for IRE specificity over other branches is not strong. It is unclear why PRKCSH would bind selectively to IRE when the luminal domains are highly homologous and in fact interchangeable. In addition, interpretation of results and quality of some key experiments needs to be re-evaluated. Specific points for consideration are listed below.

- 1.** What happens to glucosidase activity during ER stress? I assume that PRKCSH is being redirected and thus would limit activity. Alternatively, while the authors show direct activation, of this subunit, is the entire enzyme complex recruited and is glucosidase activity needed for activation of IRE?

Response: We appreciate this important comment. The ER glucosidase II (GII) enzyme is a heterodimer consisting of a catalytic subunit, GII α , and PRKCSH (termed regulatory GII β subunit) (**Protein Sci. 2016, 25:2095-2101**). PRKCSH contributes to the efficient enzymatic activity of GII owing to its two critical functions: recognition of the substrate, high-mannose-type glycans, and maintaining stability of the GII α subunit via its retention in the ER (**Mol Biol Cell. 2009, 20:3974-3984; Mol Biol Cell. 2011, 22:1810-1823**). However, to date, it has not been reported how ER stress affects the GII activity and the interaction between GII α and PRKCSH.

Following the reviewer's comment, we examined the impact of ER stress on GII activity. Generally, specific activity of GII is measured by *in vitro* assay (cell-free system) using recombinant proteins (**J Biol Chem. 2013, 23:16460-16475**). However, a method to measure GII activity under ER stress *in vivo* or *ex vivo* has not been reported yet. Thus, we performed co-immunoprecipitation using anti-GII α antibody to directly determine the effect of ER stress on the interaction between PRKCSH and GII α . In the revised manuscript, we provide new evidence that ER stress induces the dissociation of the GII complex (**Fig. 5g**; described in the Results of the revised manuscript, **page 15**; and in the Discussion, **page 20**). PRKCSH binding to GII α is required for maintaining the GII α level in the ER and GII enzymatic activity (**Protein Sci. 2016, 25: 2095-2101**); therefore, it is plausible that ER stress ultimately reduces glucosidase activity.

2. Looking at the data, it is clear that PRKCSH is not required for activation; rather it seems to provide a boost in signaling. Indeed, it appears that the kinetics of Xbp1 splicing are unaltered and differences in spliced Xbp are not apparent until 5 hours suggesting that PRKCSH modifies the response but is not necessary.

Response: Thank you for this insightful comment, with which we totally agree. This

suggestion was very helpful for interpreting clearly the role of PRKCSH in the regulation of IRE1 α activation. Thus, we corrected the sentences in the revised manuscript as suggested (modified in the Results of revised manuscript, **pages 9 and 10**).

3. With respect to the comment above, the dose of tunicamycin used throughout are very high. A dose response and use of a lower dose of TM would be more compelling throughout.

Response: Treatment with 5–10 $\mu\text{g/ml}$ tunicamycin is generally used in ER stress research. Numerous groups have used this concentration for studies of ER stress response in various cancer cells such as lung cancer or lymphoma cells (**J Cell Sci. 2016, 129:2317-2328; Oncotarget 2016, 22:76827-76839; Mol Biol Cell. 2014, 25:1411-1420; Blood 2009, 113:626-634**).

Following the reviewer's comment, we investigated changes in XBP1 splicing in L02-PRK cells treated with 1.25–10 $\mu\text{g/ml}$ tunicamycin (**Fig. 2c**; described in the Results of the revised manuscript, **page 8**). Furthermore, we determined the time dependence of the response of XBP1 splicing from 2 h to 16 h of treatment with 10 $\mu\text{g/ml}$ tunicamycin (**Fig. 2a, 2b**). In the revised manuscript, we revised the description of the results for splicing of XBP1 mRNA and protein and replaced the previously presented data with new data. According to revised Fig. 2a–c, a remarkable difference in XBP1 splicing was observed 4–8 h after addition of 10 $\mu\text{g/ml}$ tunicamycin.

4. In the beginning of the results, the authors never define what PRK cells are either in text or figure legends.

Response: According to this comment, we added a sentence describing *PRKCSH* overexpression (L02-PRK) and knockout (L02-PRK KO) in L02 normal liver cells in the

revised manuscript (in the Results, **page 8**, and Methods, **page 25**).

5. Figure 2 and 3 provide evidence for PRKCSH in increasing Xbp1 splicing and later that it increases IRE1 auto-p. However, the two are never tied together. The authors should determine whether Ire1 is necessary for PRKCSH-dependent effects on Xbp1 (knockdown of Ire or Ire1 inhibitor). It would be of equal interest to determine whether overexpression of Ire1 (which increases its activity through forced oligomerization) would overcome the effect of PRKCSH knockout or knockdown. The issues with the dose of TM, the kinetics of the response, this data would help support that the impact is directly on Ire1.

Response: We have shown the results for the splicing of XBP1 mRNA and protein, and activation of ERK1/2 MAPK in IRE1 α -knockdown L02-PRK cells (Supplementary Fig. 8 in the original manuscript), which suggest that IRE1 α is necessary for PRKCSH-mediated XBP1 splicing and MAPK activation. Considering the reviewer's comment, we moved these data from Supplementary Fig. 8 to Fig. 4 to emphasize this aspect in the revised manuscript (**Fig. 4i, 4j, 4k**; described in the Results of the revised manuscript, **pages 12–13**). Our data showed that the boosting effect of PRKCSH on XBP1 splicing and MAPK activation is almost completely abolished by IRE1 α silencing.

Following the reviewer's comment, we performed additional experiments to determine whether IRE1 α overexpression would overcome the effect of PRKCSH knockout. In the revised manuscript, we demonstrate that IRE1 α overexpression (hIRE1 α .pcD construct was provided by Dr. Randal Kaufman) increased the splicing of XBP1 mRNA and protein and activation of ERK1/2 MAPK in L02-PRK KO cells (**Fig. 4i, 4j, 4k**; described in the Results of the revised manuscript, **pages 12–13**). Taken together, our results provide clear evidence that IRE1 α is a downstream target of PRKCSH during ER stress response.

6. The specificity for Ire1 is not convincing given high homology of all UPR transducers in their luminal domains. In the supplemental data, it is not sufficient to only show PERK-p. The authors need to perform the same time course as with Ire1 and XBP1, and examine downstream signaling; p-eIF2, ATF4 and CHOP. These experiments should also be performed in PRKCSH k/o cells for clarity.

Response: According to the reviewer's comment, we performed immunoblot analysis to check the levels of PERK phosphorylation and ATF4 expression using a time course similar to that used for determining IRE1 phosphorylation in PRKCSH-overexpressing and knockdown cells. We also investigated those levels using L02-PRK KO and Huh-PRK KO cells, which were newly established during revision. We examined the expression of *Ero1B*, a downstream target gene of PERK pathway, using PRKCSH-overexpressing (L02), knockdown (Huh-7), and KO (L02 and Huh-7) cells. We provide evidence that PRKCSH is not involved in the regulation of PERK pathway activation (**Supplementary Fig. 5a–e**; described in the Results of the revised manuscript, **page 11**).

7. The title of the section "PRKCSH directly activates Ire1" is misleading and not accurate. The data demonstrate it binds, but that binding correlates with increased auto-P. However, no data demonstrate activation and PRKCSH is clearly not required for binding since there is activity in its absence.

Response: Thank you for the insightful comment. According to this, we revised the title and the sentence to explain our result clearly in the revised manuscript (described in the Results title of the revised manuscript, **pages 11 and 14**).

8. The experiments interpreted to demonstrate that PRKCSH increases oligomerization

could reflect formation of inclusions at the proteasome that reflects overexpression. These experiments do not in fact establish oligomerization. Use of differential tagging would be one possible approach to compliment and control for this issue.

Response: To investigate IRE1 α oligomerization under ER stress, many research groups have used a fluorescently tagged IRE1 α fusion construct. This tag is placed in the nonconserved linker between the transmembrane domain and the kinase domain, because the C- or N-terminally GFP-tagged constructs are not fully functional (**Proc Natl Acad Sci USA 2010, 107:16113-16118; Developmental Cell 2012, 23:1141-1152; J Biol Chem 2011, 286:30859-30866; Nature 2009, 457:736-740**). We have also used a Venus-tagged fluorescent IRE1 α fusion construct to determine the effect of PRKCSH deficiency or overexpression on IRE1 α oligomerization under ER stress. This construct has been provided by Dr. Nozomu Kono's group, who also used it for the study of IRE1 α oligomerization under ER stress (**Genes Cells 2013, 18:798-809**).

Nevertheless, to address the reviewer's concern, we examined whether the oligomerized IRE1 α is co-localized with the proteasome. In the revised manuscript, we provide clear evidence that a Venus-tagged IRE1 α fusion protein is not associated with the proteasome under either resting or ER stress conditions, excluding the possibility of inclusion formation in the proteasome by overexpressed IRE1 α (**Supplementary Fig. 6**; described in the Results of the revised manuscript, **page 12**).

9. Defining the E/P domain as necessary for binding is excellent. However, it is disappointing that the authors did not: 1) demonstrate specificity for Ire; 2) determine whether it is sufficient to activate Ire1 and do so selectively.

Response: To determine whether the interaction between PRKCSH and IRE1 α through the E/P domain is sufficient to activate IRE1 α as suggested by the reviewer, we constructed

additional PRKCSH mutants. Together with the result shown in Fig. 5e, the experiments using these mutants confirmed that the E/P domain is essential for the interaction between PRKCSH and IRE1 α (**Fig. 5f, Supplementary Fig. 7a**).

To identify the compartment in which the interaction between PRKCSH and IRE1 α occurs, we constructed a PRKCSH mutant lacking the N-terminal signal peptide (Δ S/G2B). The results provide additional evidence that the interaction between PRKCSH and IRE1 α occurs in the ER but not in the cytoplasm (**Fig. 5f**; described in the Results of the revised manuscript, **page 14**).

Subsequently, we used the Δ G2B, Δ S/G2B, and E/P mutants to determine whether the PRKCSH–IRE1 α interaction is sufficient for IRE1 α activation. Both the Δ G2B and E/P mutants interacted with IRE1 α not only under ER stress but even under resting conditions (**Fig. 5f**). Expression of these two mutants promoted both IRE1 α activation and splicing of XBP1 under ER stress (**Fig. 5i, 5j**; described in the Results of the revised manuscript, **page 15**), whereas these mutants failed to activate IRE1 α pathway under resting condition (**Fig. 5i, 5j**; described in the Results of the revised manuscript, **page 15**, and Discussion, **pages 19–20**). Thus, in the revised manuscript, we demonstrate that the interaction between the two proteins is necessary for boosting the activation of IRE1 α by ER stress, but is not sufficient for triggering its activation.

The specificity issue is addressed in the response to comment 10.

10. With regard to specificity, if PRKCSH indeed targets Ire1 specifically, interchanging luminal domains should alter or switch specificity and provide direct support for the authors hypothesis.

Response: Although the luminal domains of IRE1 α and PERK are structurally similar, their

amino acid sequences and structures are not identical (**J Cell Biol. 2004, 167:445-456**). Interchanging the luminal domains between IRE1 α and other sensors would give another possibility for demonstrating the specificity; however, it cannot reproduce the exact physiological conditions because such proteins would not be natural. Therefore, the specificity can be examined in another way, by comparing the interaction of the protein of interest with IRE1 α and other sensors (**Developmental Cell 2012, 23:1141-1152**).

Considering the reviewer's comment, we performed co-immunoprecipitation with anti-PERK antibody and found that PRKCSH does not interact with PERK under either resting or ER stress conditions, supporting the specific interaction of PRKCSH with IRE1 α (**Supplementary Fig. 5f**; described in the Results of the revised manuscript, **page 14**, and in the Discussion, **page 19**).

Since ATF6 was not detected in cell lysates prepared for co-immunoprecipitation, as shown below, we were unable to assess the interaction between ATF6 and PRKCSH. However, we think that PRKCSH specifically interacts with IRE1 α because the luminal domain of ATF6 has no structural similarity with that of IRE1 α and PRKCSH does not interact with PERK.

Reviewers' comments:

Reviewer #1 (Remarks to the Author):

The authors have responded positively to the majority of the critiques and the manuscript has been improved substantially. Specifically, the new data from analyses of TCGA databases and protein levels in of additional cell lines and samples supports the notion that SPRHCC is upregulated in multiple malignancies and correlates with expression levels of XBP-1. Moreover, the authors have generated CRISPR KO cell lines which show a clearer dependency of the UPR and cell survival on PRKCSH expression.

The heavy reliance on high doses of tunicamycin and thapsigargin for some of the key findings is still problematic; however, this is counterbalanced by the demonstration that hypoxia and low glucose can also elicit PRKCSH expression. One key unanswered point, which in my view diminishes overall impact for this work is the lack of any in vivo phenotype when PRKCSH is abrogated.

Also, as a more minor issue, the new "clonogenic survival data depicted in figure 6e, are not technically "clonogenicity" data, since the cells were plated at high density and plate area coverage rather than individual colonies are measured. Though these results do imply that PRKCSH is impacting clonogenic death, the title of this figure panel and corresponding text should be changed to "Cell survival".

The manuscript is well-written, and appropriate statistical analyses appear to have been performed. The experimental design is provided with sufficient detail so that the work should be able to be reproduced.

Overall, the work presented in this manuscript is novel and adds another layer of complexity in the regulation of the IRE1-arm of the UPR in pathological conditions.

Reviewer #2 (Remarks to the Author):

The authors have in general addressed my concerns.

Reviewers' comments:

Reviewer #1 (Remarks to the Author):

The authors have responded positively to the majority of the critiques and the manuscript has been improved substantially. Specifically, the new data from analyses of TCGA databases and protein levels in of additional cell lines and samples supports the notion that PRKCSH is upregulated in multiple malignancies and correlates with expression levels of XBP-1. Moreover, the authors have generated CRISPR KO cell lines which show a clearer dependency of the UPR and cell survival on PRKCSH expression.

The heavy reliance on high doses of tunicamycin and thapsigargin for some of the key findings is still problematic; however, this is counterbalanced by the demonstration that hypoxia and low glucose can also elicit PRKCSH expression. One key unanswered point, which in my view diminishes overall impact for this work is the lack of any *in vivo* phenotype when PRKCSH is abrogated.

Also, as a more minor issue, the new “clonogenic survival data depicted in figure 6e, are not technically “clonogenicity” data, since the cells were plated at high density and plate area coverage rather than individual colonies are measured. Though these results do imply that PRKCSH is impacting clonogenic death, the title of this figure panel and corresponding text should be changed to “Cell survival”.

The manuscript is well-written, and appropriate statistical analyses appear to have been performed. The experimental design is provided with sufficient detail so that the work should be able to be reproduced.

Overall, the work presented in this manuscript is novel and adds another layer of complexity in the regulation of the IRE1-arm of the UPR in pathological conditions.

Response: We appreciate your positive comments. Generally, xenograft mouse model is used for monitoring either *in vivo* tumor cell growth rate under physiological condition or tumor cell survival/death under treatment with cancer therapeutic agents (**Nat. Rev. Cancer. 2005, 5:876-885; Eur. J. Cancer. 2004, 40:827-836; Cell. Death. Differ. 2016, 23:1615-1627; Oncogene 2019, 38:375-389**). Under physiological condition, *in vivo* tumor growth rate only give evidence for tumor cell proliferation, similar to *in vitro* tumor cell proliferation, but may not give direct indication for the tumor survival/death (**Nat. Cell. Biol. 2009, 11:312-319; Nature 2018, 556:249-254; Oncogene. 2016, 35:2756-2765; Mol. Cell. Biol. 2007, 27:2103-2119; Oncotarget. 2014, 5:2635-2647**). As evident in Figure 6e, when there was

no stress, no significant difference was shown in *in vitro* cell growth rates between PRKCSH over-expressed L02 and control cells. The growth rate of PRKCSH-deficient Huh7 cells was also similar with control cells. There was significant difference in cell survival/death only when cells were under ER stress conditions. Thus, we think that investigating *in vivo* tumor growth of PRKCSH-deficient cells under physiological condition may show similar growth rate because this system only represents cell proliferation, not the cell survival or death.

In ER stress condition research, many studies about tumor cell survival have been performed mainly using *in vitro* models upon treatment with stress inducers (**Cell Death and Differentiation 2012, 19:1013-1026; EMBO J. 2011, 30:4465-4478**). In the present study, we also showed evidences for tumor cell survival under ER stress condition using *in vitro* models with PRKCSH knock-out and over-expressed cells. We think that the most important *in vivo* relevance of some study can be obtained with the patient data. In this regard, we showed that the level of PRKCSH is associated with multiple human tumor tissues, tumor progressions, and patient survivals. Considering that the mouse study does not always match with the human data, our results from patient data along with *in vitro* tumor survival study may represent the *in vivo* relevance on the role of PRKCSH in human tumor promotion. Thus, we think that the lack of *in vivo* mouse study will not be able to jeopardize our findings on the novel role of PRKCSH in human tumor promotion.

Finally, we have corrected “clonogenic survival” to “cell survival” in the revised manuscript as suggested (modified in the Results, Methods, and figure legends of revised manuscript, **pages 16, 32, 48, and 49**).

Reviewer #2 (Remarks to the Author):

The authors have in general addressed my concerns.

Response: Thank you for your helpful comments during the revision.

Reviewers' comments:

Reviewer #1 (Remarks to the Author):

The authors have responded positively to the majority of the critiques and the manuscript has been improved substantially. Specifically, the new data from analyses of TCGA databases and protein levels in of additional cell lines and samples supports the notion that PRKCSH is upregulated in multiple malignancies and correlates with expression levels of XBP-1. Moreover, the authors have generated CRISPR KO cell lines which show a clearer dependency of the UPR and cell survival on PRKCSH expression.

The heavy reliance on high doses of tunicamycin and thapsigargin for some of the key findings is still problematic; however, this is counterbalanced by the demonstration that hypoxia and low glucose can also elicit PRKCSH expression. One key unanswered point, which in my view diminishes overall impact for this work is the lack of any *in vivo* phenotype when PRKCSH is abrogated.

Also, as a more minor issue, the new “clonogenic survival data depicted in figure 6e, are not technically “clonogenicity” data, since the cells were plated at high density and plate area coverage rather than individual colonies are measured. Though these results do imply that PRKCSH is impacting clonogenic death, the title of this figure panel and corresponding text should be changed to “Cell survival”.

The manuscript is well-written, and appropriate statistical analyses appear to have been performed. The experimental design is provided with sufficient detail so that the work should be able to be reproduced.

Overall, the work presented in this manuscript is novel and adds another layer of complexity in the regulation of the IRE1-arm of the UPR in pathological conditions.

Response: We appreciate your positive comments. Considering reviewer’s comment, we performed the *in vivo* tumor growth analysis in xenograft nude mouse model using Huh-PRKCSH WT and KO cell lines. In the revised manuscript, we provide clear evidence that PRKCSH has a role in tumor growth under physiological stress conditions of tumor microenvironment (Fig. 6f and Supplemental Fig. 9; described in the Results of the revised manuscript, **page 16**).

Finally, we have corrected “clonogenic survival” to “cell survival” in the revised manuscript as

suggested (modified in the Results, Methods, and figure legends of revised manuscript, **pages 16, 32, 48, and 49**).

Reviewer #2 (Remarks to the Author):

The authors have in general addressed my concerns.

Response: Thank you for your helpful comments during the revision.

REVIEWERS' COMMENTS:

Reviewer #1 (Remarks to the Author):

In this revised version, the authors provide compelling data that knocking out PRKCSH in tumor cells results in significant attenuation of tumor growth. This new data, in my view, enhances substantially the biological role of the main findings. All other issues have also been adequately addressed.